# Commercial traceability of *Arapaima* spp. fisheries in the Amazon Basin: can biogeochemical tags be useful?

Luciana A. Pereira[1], Roberto V. Santos[1], Marília Hauser[2,4], Fabrice Duponchelle[3,4], Fernando Carvajal[5], Christophe Pecheyran[6], Sylvain Bérail[6], Marc Pouilly[3,4]

[1] Laboratorio de Geochronologia, Universidade de Brasília, Brasília,70910-900, Brazil
[2] Laboratory of Ichthyology and Fishery, Department of Biology, Universidade Federal de Rondônia, Porto Velho, Brazil.
[3] Institut de Recherche pour le Développement, IRD, Unité Mixte de Recherche « Biologie des Organismes et Écosystèmes Aquatiques » (UMR BOREA – MNHN, CNRS-7208, UPMC, UCBN, IRD-207), F75005 Paris, France
[4] LMI-EDIA (Laboratoire Mixte International – Evolution et Domestication de l'Ichtyofaune Amazonienne).
[5] ULRA, Universidad Mayor de San Simon, Cochabamba, Bolivia
[6] Université de Pau et des Pays de l'Adour/CNRS, LCABIE-IPREM, Pau, France

Correspondence to: Luciana A. Pereira (lualvesp.bio@gmail.com)

**Abstract.** The development of analytical tools to determine the origin of fish is useful to better understand patterns of habitat use and to monitor, manage and control fisheries, including certification of food origin. The application of isotopic procedures to study fish calcified structures (scales, vertebrae, and otoliths) may provide robust information about the fish geographic origin and environmental living conditions. In this study, we used Sr and C isotopic markers recorded in otoliths of wild and farmed commercialized pirarucu (*Arapaima* spp.) to evaluate their prediction potential to trace the fish origin. Wild and farm fish specimens, as well as food used for feeding pirarucu in captivity, were collected from different sites. Isotope analyses of otoliths performed by IRMS ($\delta^{13}$C) and LAfs-MC-ICPMS ($^{87}$Sr/$^{86}$Sr) were compared to the isotopic composition of water and of the food given to the fish in the farms. Wild fish specimens that lived in environments with the largest fluctuation of river water Sr isotope ratios over time presented the largest Sr isotope variations in otoliths. A quadratic discriminant analysis on otolith isotopic composition provided 58% of correct classification for fish production (wild and farmed) and 76% of correct classification for the fish region. Classification accuracy for region varied between 100% and 29% for the Madeira and the lower Amazon fishes, respectively. Overall, this preliminary trial is not yet fully developed to be applied as a commercial traceability tool. However, given the importance of *Arapaima* spp. for food security and the generation of economic resources for millions of people in the Amazon basin, further analyses are needed to increase the discrimination performance of these biogeographical tags.

## 1   Introduction

Food production is becoming increasingly associated with sustainable practices ensuring environmental preservation goals. Food origin and production conditions have become important issues in a national and international trade to attest of adequate practices. For instance, consumers want to know whether fish being consumed belongs to an endangered or vulnerable species and whether they have grown in natural or farmed conditions (Baffi and Trincherini, 2016; Kim et al., 2015; Pracheil

et al., 2014). In order to address some of these questions, and further improve wild and farming fish management, recent studies have used biogeochemical tracers to better understand fish population dynamics, their ecological strategies and stock origin (Brennan and Schindler, 2017; Kennedy et al., 2005; Kerr and Campana, 2013; Pracheil et al., 2014; Rojas et al., 2007; Thresher, 1999). These tools have also been used to control national and international fish trade (Pracheil et al., 2014) and to

identify the geographical origin and conditions under which fish have been raised (Barnett-Johnson et al., 2008; Bell et al., 2007; Rojas et al., 2007; Turchini et al., 2009). A sustainable community-based management is important to ensure secure and fair food production, thus valorizing the local economy, respecting the ecosystems functioning, and maintaining ecosystem services.

Determination of fish geographic origin by stable isotope analyses has been investigated by different authors aiming

to characterize food origin, manage fisheries and fishes stocks, build knowledge of species life history, identify critical habitats for conservation, and avoid overexploitation of fish stocks (Baffi and Trincherini, 2016; Duponchelle et al., 2016; Garcez et al., 2015; Hauser, 2018; Hegg et al., 2015; Jordaan et al., 2016; Pouilly et al., 2014; Pracheil et al., 2014). The methodological assumptions to discriminate fish stocks are based on (i) the isotopic heterogeneity among stocks and (ii) the low mobilization rates in the analyzed tissue (Kerr and Campana, 2013). Isotopic information can be extracted from fish calcified structures

(scale, otolith or vertebrae) in order to preserve fish integrity for the commercial use and eventually reconstruct fish life history (Campana, 1999; Pouilly et al., 2014; Pracheil et al., 2014). Strontium isotopes have been used as origin tracer of food products because of their robust response in terms of origin authenticity and fraud detection (Baffi and Trincherini, 2016). On the other hand, carbon isotopes have been used to distinguish farmed and wild fishes according to feeding patterns (Rojas et al., 2007; Turchini et al., 2009). Fish otoliths, or ear bones, are calcified structures that grow continually and record ambient condition

along fish's lives, from hatching to death (Campana 1999). Since Sr isotopes in otolith are not reabsorbed and do not fractionate during biological uptake, the isotopic ratio of this element is a robust geographic marker (Kennedy et al., 2000; Kerr and Campana, 2013; Pouilly et al., 2014). Most studies using Sr isotopes in fish otoliths were performed on marine and freshwater ecosystems of temperate regions (Comyns B. H., 2008; Gillanders, 2002; Kennedy et al., 1997, 2000, 2002; Woodhead et al., 2005). Only a few studies have focused on fish migration and living conditions in tropical river systems (Duponchelle et al.,

2016; Garcez et al., 2015; Hauser, 2018; Hegg et al., 2015; Pouilly et al., 2014; Sousa et al., 2016; Walther et al., 2011).

The Amazon basin has the largest rainforest on the planet and constitutes a complex system of rivers, lakes, and wetlands (Oliveira, 1996). It has a variety of different river waters (white, black, and clear) that drains complex and heterogeneous geologic formations (Santos et al., 2015). Because these rivers drain rocks with different origins and ages, their waters present contrasting Sr isotopic compositions, thus providing an adequate scenario for the application of Sr isotopes as

a geographic tracer (Duponchelle et al., 2016; Hauser, 2018; Pouilly et al., 2014; Santos et al., 2015). The region is also known to support a large diversity of fish species, many of which play an important economic role in the region, such as the *Arapaima* spp., known as one of the largest freshwater fish genus (Hrbek et al., 2007; Queiroz, 2000; Stone, 2007). The four described species (*A. agassizii, A. mapae, A. leptosoma,* and *A. gigas)* of this genus are endemic to the Amazon basin, where they are popularly called Pirarucu or Paiche (Arantes et al., 2010; Stewart, 2013a, 2013b). This genus is socially, economically and

ecologically important in the region because it constitutes one of the main food sources for the local community, providing important economic resources on a local and regional scales (farming, fishing, trading). Due to overexploitation, *Arapaima* spp. have been classified as vulnerable by CITES and fishing is subject to legal restriction, such as seasonal fishing prohibition, the minimum size of capture and, most important, *Arapaima*'s commercialization is restricted to fish originated from

management areas or aquaculture farms (Feio and Mendes, 2017). Paradoxically, *Arapaima* spp. are considered as exotic invasive species in the Upper Madeira watershed in Bolivia and Peru, after being introduced in the region in the 70'(Van Damme et al., 2011; Figueiredo, 2013; Miranda-Chumacero et al., 2012).

Synergic actions initiated in 1989 and involving communitarian lake management, governmental conservation policies, ONG's projects, and aquaculture production of *Arapaima* have allowed the recovery of overexploited stocks

(Figueiredo, 2013; McGrath et al., 2015). For instance, more than 500 groups have permission for fishing *Arapaima* under an annual quota stipulated by the Federal Agency (IBAMA), which have limited resources to monitor, control and regulate fishery stocks (McGrath et al., 2015). Even though the "sustainable fishery" is certified with tags in order to allow traceability, these tags are easily frauded and illegally reused. In this context, the development of an isotopic tag to track back the original precedence of *Arapaima* fishery could reinforce the actual system of traceability and combat illegal exploitation of *Arapaima*

stocks.

Some aspects of *Arapaima* spp. biology qualify this group as a model for isotopic certification. This genus is described as sedentary, although individuals migrate locally from lateral lakes to rivers during flooding pulses in order to complete their life cycle (Arantes et al., 2010; Araripe et al., 2013; Castello, 2008). Also, because this genus is air breath dependent, individuals come out the surface to breath regularly and local fishermen can estimate populations size and realize the

certification of the communitarian management. These characteristics lead to a spatial distinction between stocks of the major Amazonian regions and allow to use biogeochemical tags as a tracer of origin. Moreover, trophic patterns of *Arapaima* vary among populations and over the ontogeny from omnivorous (Watson et al. 2013) to carnivorous or piscivorous, predominantly based on C3 sources (Carvalho et al., 2018; Domingues et al., 2006; Queiroz, 2000). Because of this variation on feeding sources (Carvalho et al., 2018; Castello, 2008), a common strategy used in aquaculture and strongly recommended by

governmental manuals of *Arapaima* farming is the nutritional training with three rations during different life stages (Ono and Kehdi, 2013). As a reference, $\delta^{13}C$ values of the food used on farming activity vary between -19.3‰ and -14.9‰, thus corresponding to a diet based mostly on C4 macrophytes, corn, and soya beans.

In this study, we hypothesize that farmed and wild fishes of different sub-basin would present different $\delta^{13}C$ and $^{87}Sr/^{86}Sr$ values, depending on their food sources ($\delta^{13}C$) and geographical origins ($^{87}Sr/^{86}Sr$). Hence, the main objective is to

test whether $\delta^{13}C$ and $^{87}Sr/^{86}Sr$ measured on fish otoliths can be used as biogeochemical tags of the geographic origin and provenance (wild vs. farmed) of *Arapaima* specimens and, consequently, evaluate if these isotopes can be used as a traceability tool. We analyzed Sr and C isotopic composition of *Arapaima*'s otoliths from farmed and wild fishes proceeding from four different Amazonian regions (Madeira, Solimões, Central Amazon, Lower Amazon). Sr and C isotope data were analyzed

across transect in *Arapaima*'s otoliths in order to identify habitat change or geographic mobility during the fish lifetime. These data were also compared to the C and Sr isotopic composition of fish food supplied by the farmers.

## 2. Material and methods

### 2.1 Study area

The Amazon basin represents a dynamic and heterogeneous ecosystem extending over more the 45% of the surface area of South America. The Amazon River and its huge network of tributaries drain different geological formations (Gaillardet et al., 1997; Gibbs, 1967; Santos et al., 2015; Stallard, 1980; Stallard and Edmond, 1983) covered by primary forests, chaparral savannas, floodplains, and swamps. These habitats are therefore some of the most biodiverse in the world, particularly in regard to the Amazonian freshwater fish fauna, which are under pressure of degradation by dams building, mining, land cover and global climate change (Anderson et al., 2018; Carnicer et al., 2015; Castello et al., 2013; Castello and Macedo, 2016; Finer and Jenkins, 2012; Forsberg et al., 2017; Latrubesse et al., 2017; Lees et al., 2016; Winemiller et al., 2016).

The Amazon basin is a geomorphological depression located between two old and stable geological regions: the Guiana shield at the North; and the Brazilian shield at the South. While the Andes Mountain chain limits the western border of the basin, cratonic terrains and the Atlantic Ocean limit its eastern border. Owing to its complex geological history, rivers of the Amazon basin drain rocks with a wide range of Sr isotope compositions (Santos et al., 2015). For example, the Madeira River and Negro River drain old rock formations, such as Precambrian and Ordovician rocks that imprint a strong radiogenic Sr isotope signature in their waters (respectively $0.7168 \pm 0.0007$ and $0.7318 \pm 0.0074$, Santos et al., 2015). In contrast, the Solimões river drains younger formations as well as carbonate rocks, so that their water is characterized by less radiogenic Sr isotope ratios ($0.7091 \pm 0.0002$, Gaillardet et al., 1997; Santos et al., 2015). Because of this heterogeneity, Sr isotopes in Amazon river waters may be used as a robust biogeographic marker for aquatic fauna (Duponchelle et al., 2016; Garcez et al., 2015; Hauser, 2018; Hegg et al., 2015; Pouilly et al., 2014; Sousa et al., 2016).

The carbon isotopic composition of an organism depends primarily on the isotopic composition of the primary producer that constitutes the basis of the trophic chain it feeds on. In particular, plants present two main photosynthetic pathways, C3 and C4, which $\delta^{13}C$ range of values are contrasting (-32‰ to -24‰ for C3 plants; -14‰ to -9‰ for C4 plants; De Niro & Epstein, 1978). In general, wild Amazon fishes feed dominantly on a trophic chain derived from C3 carbon source (Araujo-Lima et al., 1986; Forsberg et al., 1993; Jepsen and Winemiller, 2007; Marshall et al., 2008; Mortillaro et al., 2015; Watson et al., 2013).

### 2.2 Fish otolith sampling

Thirty-eight otolith samples of *Arapaima* spp. were collected in different sites of four main regions (Figure 1): 22 were obtained from professional fishermen for a commercial purpose; 6 were obtained from farmers, and 10 others were collected in Manaus and Santarém markets (Table 1). The sagittae otoliths of each specimen were extracted by head dissection

after capture. Afterward, they were washed, dried, and kept until laboratory analyses. None of the fishes had precise location of capture, and the informed origin in four main regions were (Figure 1): Solimões (Mamirauá Reserve at the Solimões river), Madeira (Yata, Beni and Madre de Dios Rivers and Ariquemes farm), Central Amazon (Manacapuru farm, at Solimoes River; Itacoatiara, at Amazon River; Novo Airao at Negro River) and Lower Amazon (Santarem, Amazon river). These regions have

a complex drainage system that may include different sources of water with different Sr isotopic compositions in each site (Table 1). For example, Mamiraua Reserve in Solimoes region is located at the confluence of five major rivers, which waters range from white (lower $^{87}$Sr/$^{86}$Sr values) to black water type (higher $^{87}$Sr/$^{86}$Sr values). Similarly, Itacoatiara is located at the confluence zone of the Amazon and Madeira rivers, which also have quite distinct Sr isotopic compositions. Therefore, the lack of precise information of collection and the variability in regional water chemistry may not be exactly matched by

literature data and may be important to explain the unexpected variance of fish otolith data.

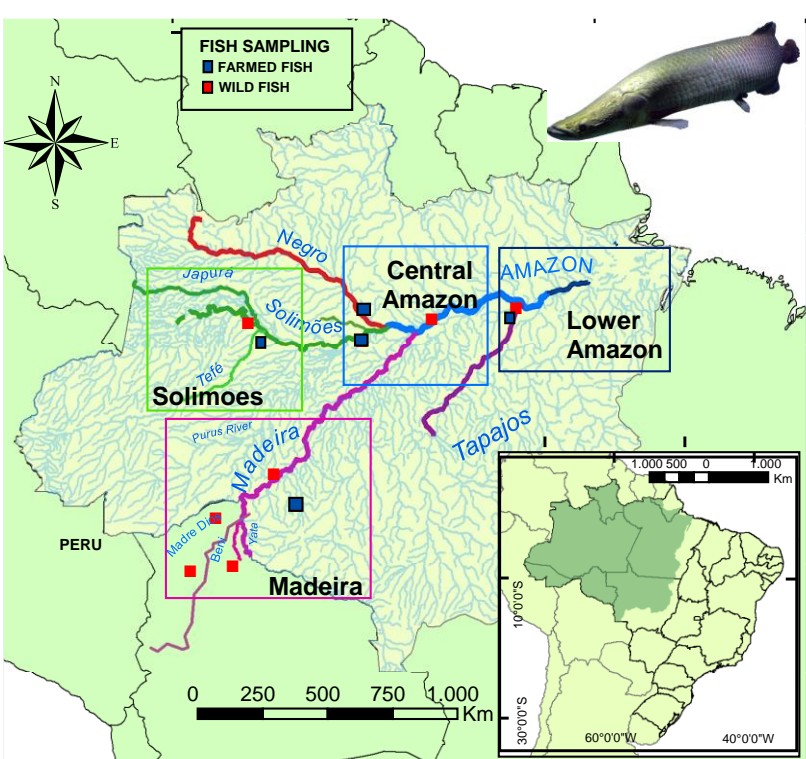

**Figure 1. Map of the Amazon basin showing the regions and sampling sites for *Arapaima* spp. Wild collect sites are represented by red squares, farm collect sites by blue squares. The Amazon River (Lower Amazon and Central Amazon) is colored blue, the**
**Solimões green, the Madeira purple, and the Negro red.**

**Table 1. δ$^{13}$C (mean ±SD) and $^{87}$Sr/$^{86}$Sr (mean ±SD) measured in sagittae otolith for 38 wild and farmed *Arapaima gigas* proceeding from four Amazonian regions with correct classification and correspondent water $^{87}$Sr/$^{86}$Sr reviewed from literature. [1]Palmer and Edmond (1992), [2]Gaillardet et al. (1997), [3]Queiroz et al. (2009), [4]Pouilly et al. (2014), [5]Santos et al. (2015) and [6]Duponchelle et al.**
**(2016)**

| Specimen code | Region | Fish origin | Site of collection | Informed origin | Mean Water $^{87}Sr/^{86}Sr$ | Otolith $^{87}Sr/^{86}Sr$ | QDA prediction | Otolith $\delta^{13}C$ |
|---|---|---|---|---|---|---|---|---|
| COO | Central Amazon | Farming | Farming | Manacapuru | 0.7091±0.0003[5,6] | 0.71543 | Central Am | -18.6 |
| I1 | Central Amazon | Wild | Fishermen | Itacoatiara | 0.7111 ±0.0004[2,5,6] | 0.71029 | Central Am | -18.3 |
| I2 | Central Amazon | Wild | Fishermen | Itacoatiara | 0.7111 ±0.0004[2,6] | 0.71267 | Central Am | -15.4 |
| I3 | Central Amazon | Wild | Fishermen | Itacoatiara | 0.7111 ±0.0004[2,6] | 0.71020 | Central Am | -20.0 |
| I4 | Central Amazon | Wild | Fishermen | Itacoatiara | 0.7111 ±0.0004[2,6] | 0.71041 | Central Am | -19.9 |
| NA1 | Central Amazon | Farming | Manaus market | Novo Airao | 0.7300 ±0.0138[2,6] | 0.70997 | Solimões | -25.2 |
| NA2 | Central Amazon | Farming | Manaus market | Novo Airao | 0.7300 ±0.0138[2,] | 0.70977 | Solimões | -25.2 |
| NA3 | Central Amazon | Farming | Manaus market | Novo Airao | 0.7300 ±0.0138[2,] | 0.70926 | Solimões | -26.0 |
| NA4 | Central Amazon | Farming | Manaus market | Novo Airao | 0.7300 ±0.0138[2,] | 0.70982 | Solimões | -24.1 |
| FMn1 | Solimões | Farming | Manaus market | Tefé | 0.7095±0.0008[2,3,6] | 0.70896 | Solimões | -25.1 |
| FMn2 | Solimões | Farming | Manaus market | Tefé | 0.7095±0.0008[2,3,6] | 0.70894 | Solimões | -24.1 |
| FMn3 | Solimões | Farming | Manaus market | Tefé | 0.7095±0.0008[2,3,6] | 0.70894 | Solimões | -23.2 |
| M1 | Solimões | Wild | Fishermen | Mamirauá Reserve | 0.7095±0.0008[2,3,6] | 0.70904 | Solimões | -25.6 |
| M2 | Solimões | Wild | Fishermen | Mamirauá Reserve | 0.7095±0.0008[2,3,6] | 0.70932 | Solimões | -25.4 |
| M3 | Solimões | Wild | Fishermen | Mamirauá Reserve | 0.7095±0.0008[2,3,6] | 0.70963 | Solimões | -24.8 |
| M4 | Solimões | Wild | Fishermen | Mamirauá Reserve | 0.7095±0.0008[2,3,6] | 0.70941 | Solimões | -25.1 |
| M5 | Solimões | Wild | Fishermen | Mamirauá Reserve | 0.7095±0.0008[2,3,6] | 0.70944 | Solimões | -25.1 |

| | | | | | | | | |
|---|---|---|---|---|---|---|---|---|
| CS1 | Lower Amazon | Farming | Santarem market | Santarem | 0.7112±0.0004[2,3,6] | 0.71382 | Lower Am | -13.7 |
| CS2 | Lower Amazon | Farming | Santarem market | Santarem | 0.7112±0.0004[2,3] | 0.71291 | Lower Am | -15.6 |
| CS5 | Lower Amazon | Farming | Santarem market | Santarem | 0.7112±0.0004[2,3,6] | 0.71479 | Lower Am | -14.0 |
| S1 | Lower Amazon | Wild | Fishermen | Santarem | 0.7112±0.0004[2,3,6] | 0.70868 | Solimões | -22.9 |
| S3 | Lower Amazon | Wild | Fishermen | Santarem | 0.7112±0.0004[2,3,6] | 0.70873 | Solimões | -26.3 |
| S4 | Lower Amazon | Wild | Fishermen | Santarem | 0.7112±0.0004[2,3,6] | 0.70868 | Solimões | -28.4 |
| S5 | Lower Amazon | Wild | Fishermen | Santarem | 0.7112±0.0004[2,3,6] | 0.70865 | Solimões | -25.6 |
| R1 | Madeira | Farming | Farming | Ariquemes | 0.7188±0.0012[1,5,6] | 0.72245 | Madeira | -8.7 |
| R2 | Madeira | Farming | Farming | Ariquemes | 0.7188±0.0012[1,5,6] | 0.71479 | Madeira | -8.3 |
| R3 | Madeira | Farming | Farming | Ariquemes | 0.7188±0.0012[1,5,6] | 0.71479 | Madeira | -4.8 |
| R4 | Madeira | Farming | Farming | Ariquemes | 0.7188±0.0012[1,5,6] | 0.71479 | Madeira | -8.6 |
| R5 | Madeira | Farming | Farming | Ariquemes | 0.7188±0.0012[1,5,6] | 0.71479 | Madeira | -8.2 |
| YA-1 | Madeira | Wild | Fishermen | Yata river | 0.7245±0.0018[4,6] | 0.72854 | Madeira | -24.8 |
| YA-12 | Madeira | Wild | Fishermen | Yata river | 0.7245±0.0018[4,6] | 0.72848 | Madeira | -26.6 |
| YA-13 | Madeira | Wild | Fishermen | Yata river | 0.7245±0.0018[4,6] | 0.72885 | Madeira | -28.1 |
| FL-1 | Madeira | Wild | Fishermen | Beni river | 0.7184±0.0011[1,6] | 0.72468 | Madeira | -27.6 |
| FL-2 | Madeira | Wild | Fishermen | Beni river | 0.7184±0.0011[1,6] | 0.72143 | Madeira | -27.0 |
| FL-20 | Madeira | Wild | Fishermen | Beni river | 0.7184±0.0011[1,6] | 0.72232 | Madeira | -24.8 |
| LV-6 | Madeira | Wild | Fishermen | Madre de Dios | 0.7119±0.0004[1,6] | 0.71441 | Madeira | -24.7 |
| LV-18 | Madeira | Wild | Fishermen | Madre de Dios | 0.7119±0.0004[1,6] | 0.71479 | Madeira | -23.7 |
| LV-20 | Madeira | Wild | Fishermen | Madre de Dios | 0.7119±0.0004[1,6] | 0.72232 | Madeira | -23.8 |

### 2.3 Samples Preparation and Analytical Methods

The otoliths were sonicated in distilled water, dried and mounted in Araldite epoxy resin at MALBEC laboratory in Montpellier University (France). Afterward, they were transversally cut with a low-speed saw (Isomed Buehler, Düsseldorf, German, 2009) to obtain a dorso-ventral slice including the otolith core. The slices were then fine-polished until the core could be seen, sonicated in distilled water and mounted on glass using crystal bond glue. Sr isotope analyses were performed at the *Laboratoire de Chimie Analytique Bio-inorganique et Environnement* (LCABIE) from the *Institut Pluridisciplinaire de Recherche sur l'Environnement et les Matériaux* (IPREM), *Université de Pau et des Pays de l'Adour* and in the laboratory PSO-IFREMER (Pole Spectrometrie Océan, Brest), France. Inter-laboratory cross-calibration was performed to confirm the

repeatability and comparability of the analysis, (see Hauser 2018 for details). The isotope ratios were measured using an fs-La-MC-ICPMS following the procedure detailed by (Claverie et al., 2009; Tabouret et al., 2010). Laser ablation conditions were 500 Hz, 20 μJ pulse energy until the depth limit ablation (<30 μm), the beam spot size of 10 μm, and velocity 5 μm/s. The Sr isotope ratios were obtained by transects (200 μm width, see Tabouret et al. 2010) along the major otolith axes, i.e. perpendicular to growth otolith lines. The laser ablated material was carried with He gas to a double torch chamber in which the laser aerosol was mixed with a 2% $HNO_3$ solution before introduction into the plasma (Barats et al., 2007). These conditions were adjusted to obtain the maximal plasma sensibility and stability. Interferent [87]Rb signal was monitored by [85]Rb, and [87]Sr/[86]Sr was corrected following Barnett-Johnson et al. (2010) procedure. Similarly, [83]Kr was measured to control [84]Kr and [86]Kr impact in [84]Sr and [88]Sr values, respectively. Finally, the ratio [86]Sr/[88]Sr was used to correct [87]Sr/[88]Sr and mass bias using the exponential law (Walther and Thorrold, 2008). Internal pattern [87]Sr/[86]Sr ratio (NIESS 22, certificated by the National Institute of Japan Environmental Studies) was analyzed at the beginning and end of each session of analysis to check the repeatability of the [87]Sr/[86]Sr measures.

Complete transects from core to edge were performed on 10 wild otolith samples, all of which presented a flat [87]Sr/[86]Sr ratio pattern along the transect. For the remaining samples, the transect was performed only on the final 1/3 part of the otoliths, which records the environmental condition during the last life period of the adult fish (Figure 2). The results presented hereafter corresponding to the final part of the otolith for all the individuals.

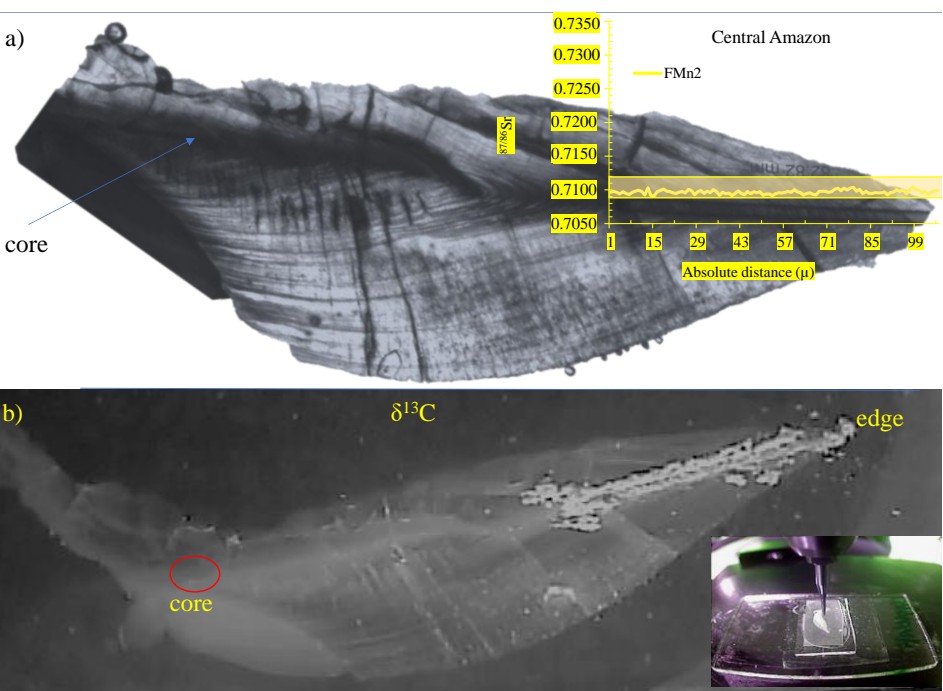

**Figure 2. Photographs of an Arapaima's otolith slice preparation. a) with the corresponding [87]Sr/[86]Sr isotopic profile over the raster, which corresponds to the final part of a core to edge transect representing the last period of life of the individual before its capture,**

analyzed by laser ablation. **The yellow rectangle illustrates the range of $^{87}Sr/^{86}Sr$ values of the Solimoes waters in the Central and Lower Amazon. b) Illustration of the micro-drilling sampling performed in the same transect in order to analyze $\delta^{13}C$ signatures.**

To obtain dietary information based on carbon isotopes, the same last 1/3 portion of each otolith was micro-drilled on the same slice preparations of otolith used to gather the Sr isotope data. The slice preparations were drilled with intervals of 6.8 mm using a New Wave Microdrill at the Universidade de Brasília. The drilled carbonate powder samples were placed directly into vials for isotope analysis. Carbon isotopes were measured using a Delta Plus V Thermo-Fisher mass spectrometer connected to a Finnigan GasBench II .at *the Laboratório de Isótopos Estáveis (LAIS), Instituto de Geociências Rede de Estudos Geocronológicos, Geodinâmicos e Ambientais (GEOCHRONOS), Universidade de Brasília, Brazil.* The results were validated against reference standards NBS 18 and 19 (respectively $\delta^{13}C$= -5.0‰ and 1.9‰).

### 2.4 Statistical Analysis

ANOVA was applied to test $^{87}Sr/^{86}Sr$ and the $\delta^{13}C$ mean difference in otoliths among 1) wild fish proceeding from the four regions, and 2) farmed fishes from the four regions. A t-test also was applied to evaluate the mean difference between all wild vs. farmed fishes. To evaluate the use of $^{87}Sr/^{86}Sr$ and $\delta^{13}C$ as a predictive tracer of fish origin (farmed or wild) and sub-basin/region of capture (Upper Amazon: Madeira and Solimões, Central Amazon, Lower Amazon), a quadratic discriminant analysis (QDA) (Anderson et al., 2010; Li et al., 2016) was carried out using a cross-validation by Jackknifed (leave one out) predictions procedure. All the statistical analyses were performed in R freeware (http://www.r-project.org/).

### 3. Results

### 3.1 Otolith Sr isotopic composition

Significant differences were observed (Figure 3) between 1) mean $^{87}Sr/^{86}Sr$ of wild fishes from the four sampled regions (ANOVA, F=18,397, p< 0,01); 2), mean $^{87}Sr/^{86}Sr$ of farmed fishes from the four sampled regions (ANOVA, F=5.614, p=0.0161), and 3) mean $^{87}Sr/^{86}Sr$ of wild vs. farmed of the same region (t = -3.764, df = 31.805, p-value p< 0,01).

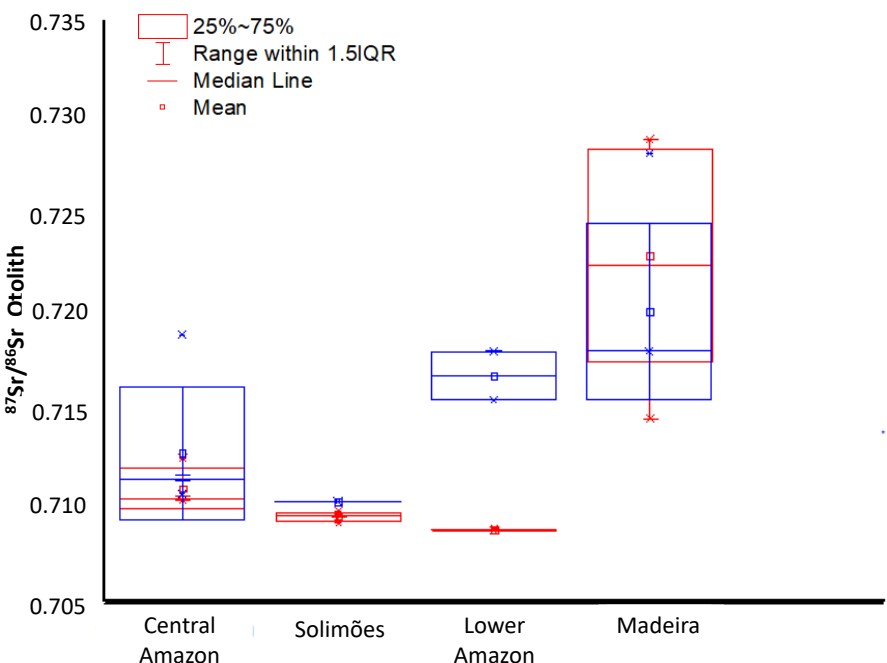

**Figure 3. Boxplot of otolith $^{87}Sr/^{86}Sr$ ratio of wild (red) and farmed (blue) *Arapaima* spp. from four Amazonian regions.**

Except for two specimens, the wild fishes presented a narrow range of $^{87}Sr/^{86}Sr$ across the otoliths (Figure 4) and their average $^{87}Sr/^{86}Sr$ values are comparable to $^{87}Sr/^{86}Sr$ of the river waters in which they were living (Table 1). The first exception is a fish from Central Amazon-Itacoatiara (I2) that exhibited $^{87}Sr/^{86}Sr$ similar to other individuals from the same site but also displayed a peak of $^{87}Sr/^{86}Sr$ value up to 0.7259 (Figure 4a). The second exception is a specimen from the Solimões-Mamirauá area (M2) that presented higher $^{87}Sr/^{86}Sr$ values (0.7223 +/- 0.0001) in comparison to the river values (0.7090-0.7100) and to other individuals from the same site (0.709-0.7110, Figure 4c).

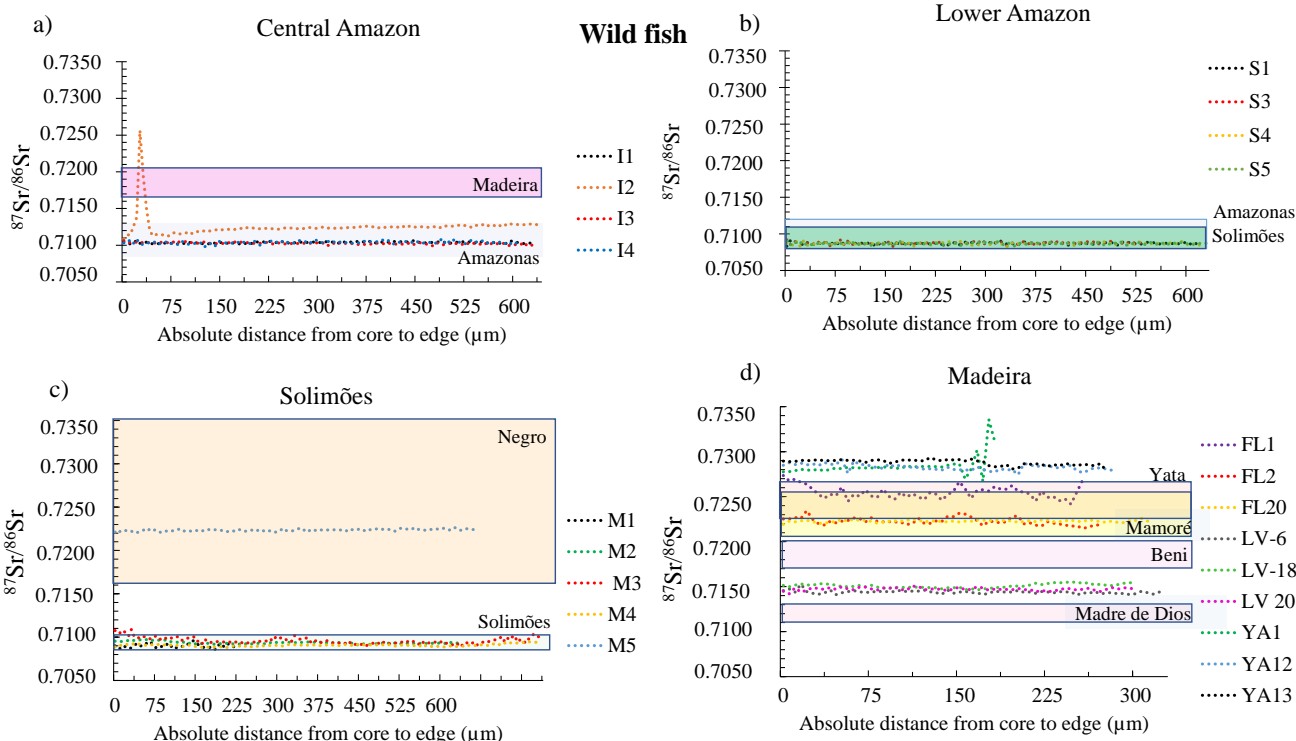

**Figure 4. Variation of $^{87}Sr/^{86}Sr$ measured by LAfs-MC-ICPMS on wild fish otoliths core - edge transects. Only the final part of the transect (approximately 1/3) is represented. Individual fishes were grouped by geographic region: a) Central Amazon; b) Madeira c) Lower Amazon and d) Solimões. The range of $^{87}Sr/^{86}Sr$ of river water dissolved matter for each geographic region is indicated by a colored rectangle (based on literature and additional analyses, see appendix 1)**

The data also revealed differences in the variability of $^{87}Sr/^{86}Sr$ values among specimens from the same region (Figure 3). Individuals from Solimões, Central Amazon (0.7090 to 0.7096) and Lower Amazon (0.7086 to 0.7087) presented a low inter-individual variation in comparison to individuals from the Upper Madeira (0.7144 to 0.7288) region.

A clear relationship also existed between the average $^{87}Sr/^{86}Sr$ values in otolith of farming fish and in local river water. Farmed fishes also generally showed a flat profile of $^{87}Sr/^{86}Sr$ values along the otolith (Figure 5), except for three individuals. One of the fishes collected directly with farmers at Manaus (COO) presented initial $^{87}Sr/^{86}Sr$ values similar to the Negro River waters. In contrast, $^{87}Sr/^{86}Sr$ values of the other specimens of the same region presented isotope ratios in the range of Solimões River waters during all their life (Figure 5a). One specimen from the Lower Amazon – Santarém area (CS1) presented important fluctuations of $^{87}Sr/^{86}Sr$ values across the otolith, corresponding to values in the range of Tapajos river, although the two other specimens collected in the same area (CS2, CS3) presented flat profiles with values intermediate between the Amazon and Tapajos river waters (Figure 5b). Finally, one of the farmed specimens of the Madeira River (R1) also showed a fluctuating $^{87}Sr/^{86}Sr$ profile (Figure 5c), although the other four fishes showed a flat and completely overlapping profile.

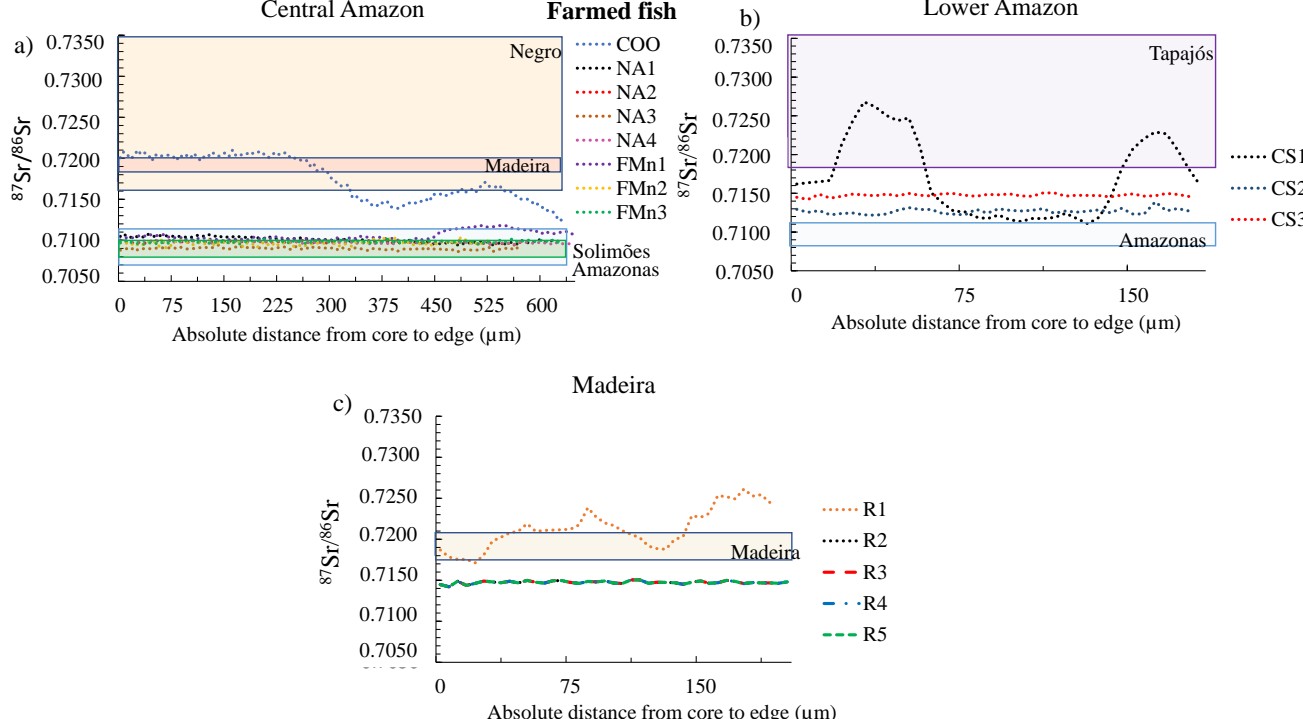

**Figure 5. Variation of $^{87}Sr/^{86}Sr$ measured by LAfs-MC-ICPMS on farmed fish otoliths core - edge transects. Only the final part of the transect (approximately 1/3) is represented. Individual fishes were grouped by geographic region: a) Central Amazon (Manaus Market); b) Lower Amazon (Santarém Market) c) Madeira (Ariquemes farm). Note that individuals R2, R3, R4, and R5 present the**
5  **same $^{87}Sr/^{86}Sr$ profile. The range of $^{87}Sr/^{86}Sr$ of river water dissolved matter for each geographic region is indicated by a colored rectangle (based on literature and additional analyses, see appendix 1).**

### 3.2 Food and Otolith Carbon isotopic composition

The $\delta^{13}C$ values of otoliths were significantly different among wild and farming specimens (ANOVA, F=124.44, p< 0,01). Most samples of wild fish present $\delta^{13}C$ consistent with C3 sources (mean -28.9 ± 1.2‰). The exceptions were all samples
10  from Itacoatiara (Central Amazon), which display a mean $\delta^{13}C$ value of -18.4‰±1.8, that fall between the C3 and C4 signatures (Figure 6). In contrast, otoliths of farmed fish presented a wide range of $\delta^{13}C$ (mean -17.1‰ ± 7.7, min = 26.0‰, max = 4.8‰). The otoliths of the fishes from the Madeira region farms presented a mean $\delta^{13}C$ value of -8.5‰ ± 0.1, indicating a strong contribution of C4 plants in their feeding source. Farmed fish from the market of Santarém (Lower Amazon) presented a mean $\delta^{13}C$ value of -14.4‰ ± 0.8 in their otoliths, thus revealing a contribution of both C3 and C4 plants in their feeding source.
15  Otoliths of farmed fish from the market of Manaus (Central Amazon) presented a lower mean $\delta^{13}C$ value (-24.7‰ ± 0.8), indicating a main C3 feeding source.

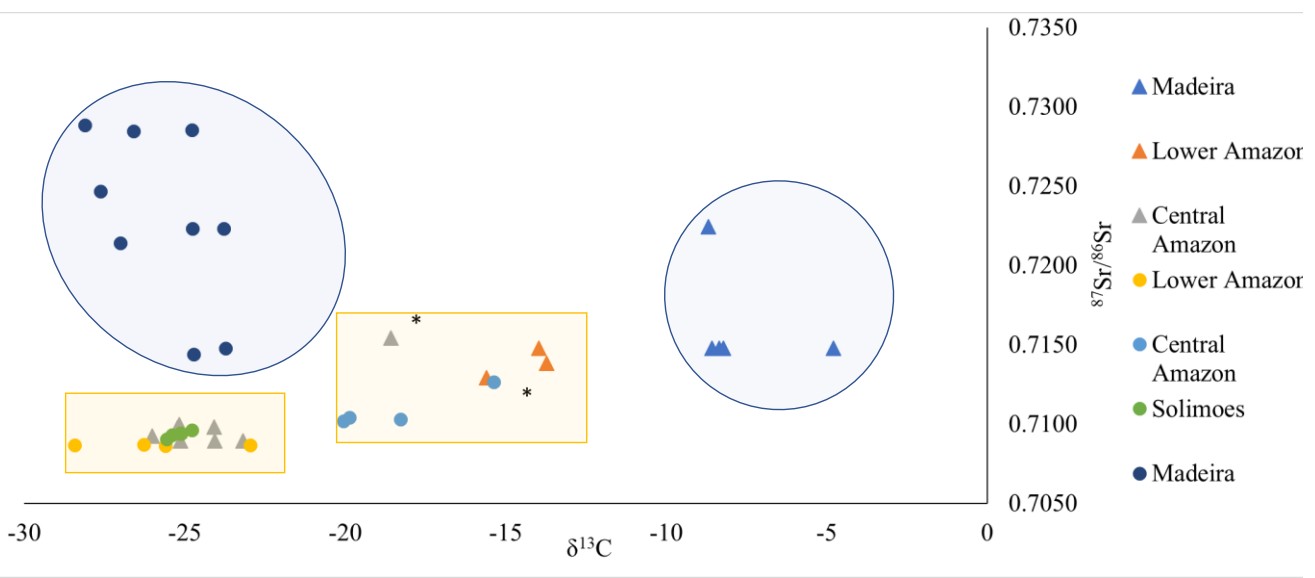

**Figure 6. Biplot of mean δ$^{13}$C and $^{87}$Sr/$^{86}$Sr for 38 wild (circle) and farmed (triangle) *Arapaima* otoliths from four geographic regions. Large blue circles represent farmed and wild fish from the Madeira, and yellow squares wild and farmed fishes from Lower Amazon, Central Amazon, and Solimões. The only two samples that are out of their group are tagged with \*.**

### 3.3 QDA Discriminant Analysis

The $^{87}$Sr/$^{86}$Sr and δ$^{13}$C isotopes biplot shows that all otolith samples fall within four main groups (Figure 6). The carbon isotopic composition combined with the average $^{87}$Sr/$^{86}$Sr ratio of the fish otolith allows to partially distinguish the different fish origin (farmed or wild) as well as their geographical region.

Wild fishes presented more negative δ$^{13}$C values (mean -24.3‰ ± 3.2) corresponding to a diet more influenced by C3 macrophytes carbon source, whereas farmed fishes presented less negative δ$^{13}$C values (mean -17.1‰ ± 7.7), corresponding to a higher influence of C4 carbon source. However, farmed fishes presented a higher variability indicating different sources of food depending on the farm. This variability led to low predictability of fish origin (58% of correct classification, Table 2).

**Table 2 Confusion matrix of fish origin classification by QDA (sample origin rows, predicted origin columns).**

|  | Farm | Wild | Correct prediction |
|---|---|---|---|
| Farm | 8 | 8 | 0.50 |
| Wild | 8 | 14 | 0.64 |
|  | Total correct prediction |  | 0.58 |

On the contrary, QDA analysis gave a higher score of correct classification of fish's region (76%, Table 3). This percentage varied between sub-basins, from 100% (Madeira) to 29% (Lower Amazon).

**Table 3 Confusion matrix of fish region classification by QDA (sample origin rows, predicted origin columns). CA = Central Amazon; LA = Lower Amazon; MA = Madeira Basin; SO = Solimoes Basin**

| | CA | LA | MA | SO | Correct prediction |
|---|---|---|---|---|---|
| CA | 10 | 1 | 1 | 0 | 0.83 |
| LA | 4 | 2 | 1 | 0 | 0.29 |
| MA | 0 | 0 | 14 | 0 | 1.00 |
| SO | 2 | 0 | 0 | 3 | 0.60 |
| Total correct prediction | | | | | 0.76 |

## 4. Discussion

The data presented showed major differences in otolith's C and Sr isotopic composition among the studied populations of *Arapaima*. In general, the $^{87}Sr/^{86}Sr$ measured in otoliths of wild specimens were similar to the $^{87}Sr/^{86}Sr$ reported in the dissolved fraction of the river water in which they were caught. Furthermore, the pattern of variation of the $^{87}Sr/^{86}Sr$ ratio along the life of these individuals was mostly flat, indicating that studied *Arapaima* predominantly stayed in water with the same chemical composition (Araripe et al., 2013; Castello, 2004, 2008; Hermann et al., 2016; Núñez-Rodríguez et al., 2015; Queiroz, 2000; Viana et al., 2007). However, some variations were recorded and are related to the lack of precise information of the fishery site and the intricate mosaic of the Amazon water chemistry. They may either correspond to the seasonal variation of water composition, or to movements between the river and adjacent lagoons, which have been shown to have higher $^{87}Sr/^{86}Sr$ values than the river (Pouilly et al. 2014).

On the other hand, most farmed fishes also presented a flat profile, but some specimens presented abrupt variations of $^{87}Sr/^{86}Sr$ values along the otolith. These variations may be produced by changes in the water isotopic composition, due for example to a transfer of the fish to another pond, or to a modification of the water source filling the pond. However, the variation could also be a consequence of food change. Most strontium otolith studies indicated the role of ambient water in the control of strontium uptake. Controversially, the role of food in the strontium otolith uptake is debated, as revealed by Sturrock et al. (2012) that reviewed the significance of food up taking processes on the Sr isotopic composition of fish. For example, Kennedy et al. (2000) suggested that food consumption in adult hatchery-reared salmon is preponderant in the uptake of strontium, although Walther and Thorrold (2006) indicated that water chemistry is the dominant factor for marine fishes. Finally, recent advances highlighted that physiological factors may also contribute to Sr control in the otolith, because these elements are transferred into the blood plasma *via* branchial or intestinal uptake, before reaching the endolymph fluid, and finally the otolith (Payan et al., 2004; Sturrock et al., 2014). Although the importance of food in strontium uptake is not clear, we may speculate that in natural condition strontium composition of fish and food sources are in equilibrium with river water, so that fish and water are directly correlated, with no significant relative contribution of food in the uptake. In artificial

condition, however, strontium composition of exogenous food source could be different from water and may result in a gap between fish and water strontium composition corresponding to the relative importance of food in the uptake.

### 4.1 Isotope record of wild *Arapaima*

The relationship between Sr isotope ratios in water and fish otoliths or scales has revealed to be a robust tool to study fish migration and geographical origin of population in the Amazon basin (Duponchelle et al., 2016; Garcez et al., 2015; Hauser, 2018; Hegg et al., 2015; Pouilly et al., 2014; Sousa et al., 2016). The $^{87}$Sr/$^{86}$Sr values of wild fishes from the Lower and Central Amazon and from the Upper Madeira and Solimões rivers presented differences according to the regions (Figures 3 and 4). Fishes from Lower and Central Amazon presented the narrowest variation of $^{87}$Sr/$^{86}$Sr values during their life. These

values were around 0.7100, which agrees with the reported values for this river waters (Santos et al. 2015). One fish presented a peak of the high $^{87}$Sr/$^{86}$Sr value of >0.7250 (Figure 4a), which could correspond to a period during which this fish has lived in a habitat with water $^{87}$Sr/$^{86}$Sr values close to the Madeira River waters or some granitic shield tributaries. Because of the sharpness of the peak, it could also be interpreted as an irregularity in the otolith. In contrast, fishes from the Upper Madeira region, including the Beni, Mamoré, Yata, and Madre de Dios rivers, presented a higher degree of variability in $^{87}$Sr/$^{86}$Sr values

(Figure 4d, from 0.7150 to 0.7350). Fishes from this region also presented a higher $^{87}$Sr/$^{86}$Sr variation across each otolith profile when compared to fishes from other sites, which is consistent with the natural seasonal variation of these older geological regions. In the Madeira waters and its upper tributaries, Santos et al (2015) observed dissolved $^{87}$Sr/$^{86}$Sr data with a high seasonal variation, owing to the nature of rocks being eroded during the rainy and dry seasons. The data presented here suggest that fish otolith also record these seasonal variations.

The wild fishes from the Solimões River were caught in the Mamirauá Reserve. This reserve does not correspond to the main channel of the Solimões channel, but to lateral lakes that developed in a mixing zone with other tributaries, some of which may be of black waters, that generally present $^{87}$Sr/$^{86}$Sr similar to that reported for the Negro river (Santos et al., 2015). One of the five fishes analyzed presented higher $^{87}$Sr/$^{86}$Sr values (>0.720), suggesting it may have lived part of its life in such a black water tributary.

Besides this regional variation, the data presented revealed that in general individual fish lived in waters with a limited range of $^{87}$Sr/$^{86}$Sr values, suggesting a resident behavior. The pattern of $^{87}$Sr/$^{86}$Sr along the otolith from the Madeira fishes presented higher variations. This could either result from movements between habitats with contrasted $^{87}$Sr/$^{86}$Sr water signatures (e.g. adjacent lakes and lagoon, as shown by Pouilly et al. 2014 for the Beni River) or from the integration of the important natural seasonal variations of $^{87}$Sr/$^{86}$Sr signature in the Madeira waters described by Santos et al. (2015).

Previous studies also concluded to a resident behaviour of *Arapaima* species (Araripe et al., 2013; Castello, 2004, 2008; Hermann et al., 2016; Queiroz, 2000; Viana et al., 2007), including a study of individual behaviour of restocked and wild *Arapaima* using radio telemetry (Núñez-Rodríguez et al., 2015). A flat $^{87}$Sr/$^{86}$Sr profile along the otolith does not directly implicate an absence of movement. Indeed, if a fish moves across two habitats presenting the same isotopic signature, the movement would not be revealed by otolith microchemistry analyses. On the other hand, Castello (2008) demonstrated lateral

migration of Pirarucu between the Solimões River and the floodplain during water pulse in the Mamirauá reserve with other observation methods. Based on our results, we can conclude that studied *Arapaima* didn't show movements across contrasted habitat (like for example white vs. black water systems, Santos et al. 2015). We cannot, however, exclude lateral movements across habitats with similar water signatures. We argue that the $^{87}$Sr/$^{86}$Sr variations observed in each otolith can be a

combination of (1) small changes in the isotopic composition of water due to diverse tributary sources in the hydrological seasonal cycle and/or (2) a lateral migration. As an example, the M3 fish from the Mamiraua showed ripples that might be interpreted as lateral movements between the Solimões white waters and adjacent lagoons or lakes with slightly higher signature (see Pouilly et al. 2014). Due to the weakness of the pattern and the absence of $^{87}$Sr/$^{86}$Sr seasonal data from lakes and rivers in the Amazon basin, more detailed studies would be necessary to confirm one or the other hypothesis of movement

behavior, which are probably complementary.

Strontium and carbon isotopes in fish otolith record different parameters during specimen life. As $^{87}$Sr/$^{86}$Sr could be used as a robust fish geographical indicator, even in small scales (Pouilly et al., 2014), carbon isotopes composition ($\delta^{13}$C) are related to the feeding source of the fish.

Most wild fishes analyzed presented $\delta^{13}$C values between -24‰ and -30‰ (Figure 6). Hence, wild *Arapaima* in this study

had $\delta^{13}$C mostly derived from C3 plants (-28.9 ± 1.2‰), as also observed in previous studies (Domingues et al., 2006; Forsberg et al., 1993; Watson et al., 2013). However, a higher contribution of C4 plants could be observed in some specimens, in particular, fishes from the Central Amazon region (Itacoatiara site) that presented more positive $\delta^{13}$C values (-15.4‰ and -20.1‰). Forsberg et al. (1993) showed that C3 may account for 82.4% to 97.5% of the fish diet in the Amazon basin. Nonetheless, it is worth mentioning that these studies analyzed muscle tissue and there may exist isotopic differences between

carbon in muscle tissue and in bone structures because of physiologic pathway incorporation. Although this fractionation is not known in Amazonian fishes, available data from Atlantic cod indicate that $\delta^{13}$C value can be 15.9‰ higher in otoliths than in body tissues (Radtke et al., 1996). If this same fractionation were applied to the otoliths of *Arapaima*, the carbon source would have an unlikely value of -44.8‰. This fractionation difference is likely lower for Amazonian fish as also suggested by the carbon isotopic composition of other calcified tissues, like scales, reported by Domingues et al. (2006). They showed that

these calcified tissues have $\delta^{13}$C values between -18.0‰ and -29.2‰, which are in the same range as samples from the present study. The more positive $\delta^{13}$C values observed in fishes from Itacoatiara in Central Amazon may reflect environmental heterogeneity related to water types (white, black and clear), channels formations in dry season, and other hydrologic seasonality related to the Flood Pulse Concept (Domingues et al., 2006; Junk et al., 1989; Oliveira et al., 2006). Moreover, these isotopic values may be related to seasonal resource availability, such as *Schizodon fasciatus* that presents major

digestibility of C4 macrophytes in the varzea areas (Forsberg et al., 1993; Mortillaro et al., 2015; Oliveira et al., 2006).

## 4.2 Isotopic variations in farmed fish otolith

In general, farmed fishes also presented a flat $^{87}Sr/^{86}Sr$ profile, except for three fishes (CS1, FMn2, and R1) that showed a larger $^{87}Sr/^{86}Sr$ profile variation when compared with wild fishes from the same region (Figure 5). These variations could be related to water's pond physical conditions in which they have been raised and/or to abrupt changes in water type, such as seasonal pond transfer. We argue that these fishes were probably used as breeders and that the changes in Sr isotope ratio indicate that they were transferred between different ponds with different Sr isotope compositions. Indeed, fish farming manuals indicate that changing the breeders from one pond to another is an important strategy to increase reproduction (Ono & Kehdi, 2013; SEBRAE, 2010). On the other hand, four fish (R2, R3, R4, and R5) of the Madeira farm had the exact same Sr isotopic profile, suggesting they have lived the last part of their lives in the same common pond.

Compared to wild fish, farmed fish also showed a higher variation of $\delta^{13}C$, thus indicating more diversified food sources. Fishes from the Madeira (Ariquemes farm) presented less negative $\delta^{13}C$ values (from -8.7‰ to -4.8‰ ), which could be related to C4-based food (DeNiro and Epstein, 1978; Sant'Ana et al., 2010), probably containing a large proportion of corn. Farmed fishes from the Lower Amazon (Santarém farms) presented intermediate values (from -15.6‰ to -13.7‰) and those from the central Amazon (Manaus farms) had more negative values (from -26.0‰ to -23.2‰) more related to C3-based food. Therefore, we conclude that our hypothesis of an artificial alimentation based on C4 plants is not always verified, and that food farming seems to depend from local or regional production or from feeding strategies used by the farm.

## 4.3 Combining $^{87}Sr/^{86}Sr$ and $\delta^{13}C$ signatures

We aimed at verifying if the combination of Sr and C isotopes may be a powerful tool to distinguish between farmed and wild specimens from different Amazonian regions. The quality of information concerning fish origin is an important parameter for sustainable fish commercialization. The absence of precise fishing site and the heterogeneity of the water system may bring uncertainties to the data but does not compromise the scope of this study and our conclusions. This is also a sensitive question considering that commercialization of *Arapaima* is only allowed from farming or management areas. In this sense, the isotope tool applied in this study can be improved to better control its commerce in the actual system of traceability by tracing back the origin of fish and combat the illegal reuse of tags on illicit fisheries.

The QDA analyses presented in our study gave the proportion of correct indication of the origin of fishes (production method: farm or wild, geographic regions). The results showed a good but not sufficient enough (>75%) proportion of correct classification of the geographic origin (mainly based on $^{87}Sr/^{86}Sr$ values). This percentage is downgraded by the overlaps of $^{87}Sr/^{86}Sr$ values of some regions (Solimoes, Central and Lower Amazon). The lack of contrast in $^{87}Sr/^{86}Sr$ between Lower Amazon, Central Amazon, and Solimões regions leads to a higher confusion: four fishes from Lower Amazon (on a total of 7) and two fishes from Solimões (on a total of 5) were misclassified in Central Amazon region, most of them prevenient from farmed sources. On the contrary, it is upgraded by some clear contrasts existing in different Amazonian sub-basins, such as the Madeira, but we can also indicate the Tapajos or Negro rivers that also presented specific values (Santos et al., 2015; review in Hauser, 2018).

On the other hand, results showed low predictability (58%) of fish origin (farmed or wild). This is mainly due to the variety of food sources used to feed the farmed fishes. We hypothesized that farms used food based on a mixture of C3 and C4 plants (soya bean, corn) but some farms apparently used food based on C3 plants, generating confusion with the food of wild fishes. On the other hand, all fishes sold in Manaus marked as farmed fishes presented C3-based $\delta^{13}C$ signatures. This could mean that these supposedly farmed-fish actually came from wild provenance, which is illegal and contributes to the over exploration of this natural resource. False information on the fish provenance would also hamper the precision of our approach.

As a preliminary intent, the method gave some interesting results that emphasize the potential of such analyses to obtain a performing tool. In only a few cases the $^{87}Sr/^{86}Sr$ values recorded in wild fish otoliths were not in agreement with the water $^{87}Sr/^{86}Sr$ of the reported origin. For instance, the $^{87}Sr/^{86}Sr$ values of wild *Arapaima* obtain from Santarém market, lower Amazon, were similar to those observed in the Solimões River (CS3). Hence, it is possible that these wild specimens were caught in the Solimões River (e.g. Mamirauá Reserve) and not in the Santarém area as reported by the fish seller. Nonetheless, because of the scarcity of water $^{87}Sr/^{86}Sr$ baseline in this area, a Santarém origin cannot be completely ruled out. Some farmed fishes may also have $^{87}Sr/^{86}Sr$ that is not in agreement with the expected values of the reported origin. For example, farmed fishes from the lower Amazon (Santarém) were probably raised in a pond filled with water from both the Amazon and Tapajós River. Thus, the farming conditions are likely to interfere with the two tracers used in this study.

## 5. Conclusion

The expected differences of $\delta^{13}C$ between farmed and wild fishes (related to artificial vs. natural food sources) could not be confirmed, owing mainly to the C4 macrophyte contribution to the natural alimentation of wild fishes and to the use of C3-based food sources for farmed fishes. False information on the fish provenance in markets may also have contributed to decreasing the precision of the approach and market sampling should be avoided in future studies. Another weakness of our approach is to the $^{87}Sr/^{86}Sr$ overlapping among Amazon sub-basins and the lack of a more extensive $^{87}Sr/^{86}Sr$ water baseline. Hence, this preliminary result is not yet fully sufficient to be applied as a commercial traceability tool and further analyses are needed to increase the discrimination performance because millions of people rely on *Arapaima* spp. to subsistence and income. Nonetheless, these initial results encourage a more detailed seasonal $^{87}Sr/^{86}Sr$ water sampling in lakes and rivers in all the four regions analyzed, and especially in the Madeira and in the Mamirauá reserve, in order to refine the spatial water base and consequently to understand the causes of the otolith profile variation in wild *Arapaima* spp. They also suggest further axes of the investigation, such as controlled physiological experiment to clarify the sources (water and food) for $^{87}Sr/^{86}Sr$ otolith assimilation pathway in farmed conditions and investigating the actual importance of C4 macrophyte influence to both farmed and wild *Arapaima* according to seasons.

## Appendix A. Water sampling and respective dissolved $^{87}Sr/^{86}Sr$.

| Local sampling | Latitude | Longitude | $^{87}Sr/^{86}Sr$ |
| --- | --- | --- | --- |

| | | | |
|---|---|---|---|
| Manacapuru | 3°17.383'S | 60°37.914'W | 0.7091+/-1 |
| Itacoatiara | 2°48.115'S | 57°56.085'W | 0.71018+/-1 |
| Novo Airão | 2° 39.688'S | 60° 53.032'W | 0.7091+/-1 |
| Mamirauá | 2° 58.164'S | 64° 53.911'W | 0.7104+/-1 |
| Tefé lake/ Solimões/ Mamirauá | 3°20.815'S | 64°42.826'W | 0.71053+/-1 |
| Tucuxi lake /Japurá/Mamirauá | 2°49.491'S | 65°00.818'W | 0.70860+/-1 |
| Japurá/Mamirauá | 2°52.614'S | 64°55184W | 0.70874+/-1 |
| Japurá mix with Aranapu/ Mamirauá | 2°14.897S | 65°11.149'W | 0.70993+/-2 |
| Aranapu/Mamirauá | 2°22.730'S | 65°15.426'W | 0.70857+/-1 |
| Solimões/Mamirauá | 2°14.897S | 65°11.147'W | 0.70858+/-1 |
| Arapaima lake/ Solimões | 2°59.016S | 64°55.193'W | 0.7088+/-2 |
| Santarém | 2° 24.212'S | 54° 44.149'W | 0.71114+/-1 |
| Santarém | 2°23.663'ST | 54°43.468'W | 0.71073+/-1 |
| Porto Velho | 8° 42.619'S | 63° 55.425'W | 0.7168+/-1 |
| Ariquemes | 9° 53.682'S | 63° 3.977'W | 0.72961+/-1 |
| Beni | 11°01.276' | 66°06.462'W | 0.71903+/-1 |
| Beni+ Madre Dios | 10°59.191'S | 66°03.440'W | 0.71310+/-1 |
| Mamoré | 14°52.982S' | 65°01.963'W | 0.72135+/-1 |

**Author contribution**

In this work, Marc Pouilly and Roberto V. Santos designed the overall project ideas; Roberto V. Santos and Luciana A. Pereira developed the sampling plan. In the project execution, Luciana A Pereira collected samples from Central and Lower Amazon and Fernando Carvajal and Marilia Hauser sampled in Bolivia and Madeira. Then, the sample preparation, $\delta^{13}$C otolith analysis, and $^{87}$Sr/$^{86}$Sr water analysis were made by Luciana A. Pereira with the supervision of Roberto V. Santos while, the $^{87}$Sr/$^{86}$Sr otolith analysis was performed by Marilia Hauser, Christophe Pecheyran and Sylvain Bérail with the supervision of Marc Pouilly and Fabrice Duponchelle. Later, Marc Pouilly and Luciana A. Pereira performed statistical and data analysis. Finally, Luciana A. Pereira prepared the manuscript with the contributions from all coauthors.

**Competing interests**

The authors declare that they have no conflict of interest.

**Acknowledgments**

This manuscript is the result of Luciana Alves Pereira master's degree in the Graduate Program in Ecology at the University of Brasília. The authors would like to thanks CNPq (Conselho Nacional de Desenvolvimento Científico e tecnológico) for the financial support to RVS (310641/2014-4 and 428843/2016-6), The University of Brasília for the financial publication support, the Laboratory of Ichthyology and Fishery at Universidade Federal de Rondônia, Porto Velho, the

Mamirauá Institute for the logistical support on field sampling, the IRD (Institute pour Recherche et Development), the Université de Pau et des Pays de l'Adour/CNRS,LIABLE-IPREM, Pau, France, the Geochronology and Isotope Geochemistry Laboratory of University of Brasilia,  and Wikus Jordan and an anonymous referee for the review and inputs for this manuscript.

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
