# Peer review of "Commercial traceability of *Arapaima* spp. fisheries in the Amazon Basin: can biogeochemical tags be useful?"

_Biogeosciences, 2018_

## Referee Comment (RC1) · Jordaan (Referee) · 12 Dec 2018

bg-2018-471 Commercial traceability of Arapaima spp. fisheries in the Amazon Basin: can biogeochemical tags be useful?

Comments

The manuscript is well written, has a proper and logical layout and makes a valid and valuable scientific contribution. The text can however benefit from minor editing (see notes below under editing).

The authors investigated a very complex river system draining an even more complex

[Figure]

geological setting. Sr isotope analyses is a good method for investigating such a system as it is mainly determined by the weathering of up-stream geology which is not expected to vary much within a 10 or even 50 year period. Please see: Jordaan, L.J., Wepener, W. and Huizenga, J.M. (2016). The strontium isotope distribution in water and fish within major South African catchments. Water SA, 42(2), 213-224. In this case very similar data was obtained in a smaller and more controlled river system but it confirms the underlying assumptions made by the authors for the Amazon system. Data gathering can be much expanded over a multi-year sampling period to include seasonal variation, if a further study is ever undertaken.

Using C isotopes is a good approach for this problem. It has however several more factors influencing isotope fractionation than Sr isotopes and should be used within the constraints of the technique, as the authors rightfully did. Data can be much expanded if a further study is ever undertaken.

The problem of obtaining fish samples with a known origin is not unique to this study and it is recommended that fish obtained from markets be treated with caution.

The value of this work lies in the fact that it can be extended to solve more than one issue. The biggest being the illegal use of protected natural fish populations. The techniques will work for other fish species as well and it will provide data as to the sediment load and erotional patterns of such large rivers.

Recommendation

The manuscript is recommended for publication.

Editing

Page 2 line 9: replace "certificate" with "characterize" Page 3 line 17: replace with: "The Amazon basin represents a dynamic and heterogeneous ecosystem extending over more the 45% of the surface area of South America." Page 3 line 18: replace: "geologic" with "geological" Page 3 line 20: replace with "These habitats are therefore some of the most biodiverse in the world, particularly in regard to the Amazonian freshwater fish fauna which are under pressure of degradation by dams, buildings, mining, land cover and global climate change" Page 6 line 12: replace "Inter laboratorian" with "Inter-laboratory" Page 6 line 13: replace "reapitability" with "repeatability" Page 7 Figure 2: replace "87/86Sr" with "87Sr/86Sr" (numbers in superscript) and use this notation consistent throughout entire document. Page 7 line 10: replace "The slice preparations were drilled per 6.8 mm interval" with "The slice preparations were drilled a 6.8 mm intervals" Page 7 line 18: replace "Also a test t was applied" with "A t-test was also applied" Page 9 Figure 4: replace "87/86Sr" with "87Sr/86Sr" (numbers in superscript) and use this notation consistent throughout entire document. Insert "87Sr/86Sr" (numbers in superscript) for part (b) of figure also. Page 9 line 8: replace "interindividual" with "inter-individual" Page 10 Figure 5: replace "87/86Sr" with "87Sr/86Sr" (numbers in superscript) and use this notation consistent throughout entire document. Insert "87Sr/86Sr" (numbers in superscript) for Central Amazon part of figure also. Page 11 Figure 6: replace "87Sr/86Sr" with "87Sr/86Sr" (numbers in superscript) Page 11 line 16: replace "(sample origin in a row, predicted origin in the column)" with "(sample origin rows, predicted origin columns)" Page 12 line 4: replace "(sample origin in a row, predicted origin in the column)" with "(sample origin rows, predicted origin columns)" Page 12 line 19: meaning unclear, please re-write sentence: "The role of food is more controversial as revealed by Sturrock et al. (2012) that reviewed significant or non-significant food up taking processes on the Sr isotopic composition of fish." Page 13 line 21: please confirm: "black water" or back water" and be consistent throughout document.

Dr. L.J. Jordaan Council for Geoscience South Africa 2018/12/12

---

## Referee Comment (RC2) · Anonymous Referee #2 · 10 Jan 2019

I have reviewed this ms and find it to deserve publication after a few revisions are made. The ms tests the general idea that biochemical tags can be used to identify the origin of harvested individuals of arapaima in various regions of the Amazon, and this can be used to improve the management of this economically important but overexploited fish. While the authors have done an apparent good job in analyzing data, I feel like the true contribution of this ms is not reflected in the text.

The introduction generally sets out the research question clearly, but there are important issues that were not considered and would help sharpen it and increase the value of this research. For instance, about 3/4 of the introduction is devoted to describe the

use of biochemical tags to trace the origin of fish worldwide, and the Amazon is introduced only after that. When the subject of the Amazon is introduced, a key idea that is missing here is spatial heterogeneity in the chemistry of river waters. That heterogeneity is what allows the authors to text the main hypothesis — yet it is not described here, not even briefly. The hypothesis only makes sense IF there is spatial heterogeneity in chemistry, so this needs to be established in the introduction (and could be expanded in methods). Also, the study focuses on arapaima and its conservation. But while a lot of space in the introduction is devoted to review the use of chemical markers for sustainable fisheries management in general terms (paragraphs 1-3), there is almost no mention of details about the conservation measures that currently are bringing arapaima back from overexploitation. Given the paper focuses on arapaima, that seems to need attention. In particular, some 500 fishing communities in the State of Amazonas in Brazil (alone, and more now in Pará State) are setting fishing quotas and selling their "sustainable" fish to the market while fulfilling strict government limits. Each individual fish harvested under this management system receives a unique, government-issued, identifying tag that buyers can use to know where the fish came from, where and when it was harvested. But many such tags are illegally re-used to allow the "legal" sale of unsustainably harvested fish. This presents a major management problem for arapaima that the study in question can help solve, because its results can potentially be used to 'trace' back the origin of the fish and hence determine if the origin of the fish matches the tag. This study should link its results to such major ongoing management initiative for arapaima. Finally, the hypothesis of the study only makes sense IF arapaima are not highly migratory and move between and among river systems with different water chemistries. As such, known data on the general migratory behavior of arapaima should be determined 'before' the hypothesis for the hypothesis to make sense. When this is done, typical habitat and food sources (some of which are presented in methods), should also be presented here, to provide context for the hypothesis. To implement such changes, I suggest shortening the first 3 paragraphs and expanding the remainder of the introduction.

The methods section is mostly okay, but a couple of things require work. Fig 1 needs to be edited so the font is readable at half-page width size; currently the most important information cannot be read for having font too small. A lot of space on the sides is used to show regions of no interest. The fig could be "zoomed in" to the area of interest. As for the analysis, there is a major mismatch in geographical precision of the otolith data. The fish otoliths were collected from fishermen residing in sites (Table 1). Each of the 'sites' mentioned, such as Itacoatiara, Manaus, or Mamirauá reserves are, in fact, enormous regions, each of which encompass several different habitats, each of which can have varied water chemistries. For instance, the Mamiráua Reserve is flooded by some five or more major river tributaries, and includes surrounding areas influenced by blackwater ria lakes. This variability in regional water chemistry is not matched by the literature data used for each "site" It also constitutes a limitation of the analytical approach undertaken. Could this lack of specificity in the otolith origin data hep explain part of the unexplained variance in the analyses? I would seem so. As such, the "match" between the otolith and water chemistry data should be presented and discussed in methods, as well as in the discussion. It does not invalidate the analysis but it add more nuance and probably helps explain its results.

Discussion: Line 10: are there movement studies showing arapaima fo not migrate long distances? If so, this is the place to cite them (again, after introducing them in the intro) Lines 12-14: this is where a well-developed discussion of the potential for lack of geographical specificity in the otolith data to influence the results could go.

In general, the text of the discussion is sound. But I find it to be too long and unclear at times, so I suggest condensing it and revising it for clarity. What is really missing is linking the results to their application, following the idea suggested above.

Spp is not italicized Line 25: use 'developed' instead of 'satisfying'

---

## Author Comment (AC1) · 5 Feb 2019

Jordaan Comments (1) comments from Referees The manuscript is well written, has a proper and logical layout and makes a valid and valuable scientific contribution. The text can, however, benefit from minor editing (see notes below under editing). The authors investigated a very complex river system draining an even more complex geological setting. Sr isotope analyses is a good method for investigating such a system as it is mainly determined by the weathering of up-stream geology which is not expected to vary much within a 10 or even 50 year period. Please see Jordaan, L.J., Wepener, W. and Huizenga, J.M. (2016). The strontium isotope distribution in water and fish within

major South African catchments. Water SA, 42(2), 213-224. In this case, very similar data were obtained in a smaller and more controlled river system but it confirms the underlying assumptions made by the authors for the Amazon system. Data gathering can be much expanded over a multi-year sampling period to include seasonal variation if a further study is ever undertaken.

(2) author's response We fully agree that it is important to monitor river seasonal variation of Sr isotope to improve the Amazon system knowledge. The Hybam project has been providing pioneer monitoring and characterization of the Amazon system, including monthly water sampling for more than 10 years at 15 sites (Santos et al. 2015). This data set constitutes a baseline and background that has benefited several studies related to fishery sciences (migration pathways, fish stock localization, or like in our manuscript commercial traceability) as well as to the evolution of the basin. These data clearly highlight the Sr isotopic composition contrast among major Amazon sub-basins as well as seasonal isotopic fluctuations related differential erosion sources (see Santos et al. 2015). Due to the logistic complexity of water sampling in the Amazon system, it is difficult and expensive to expand the monitoring although it is surely an important issue. We hope that our work to provide a new application of this knowledge, thus reinforcing the argument to improve such a monitoring program.

Our finds are in agreement with Jordaan et al. (2016), which will be included in our revised manuscript. In the Amazon basin, Pouilly et al. (2014) established the correlation between Sr isotopic ratio in water and in fish, as provided by Jordaan et al. (2016) in South Africa.

(1) comments from Referees

Using C isotopes is a good approach for this problem. It has however several more factors influencing isotope fractionation than Sr isotopes and should be used within the constraints of the technique, as the authors rightfully did. Data can be much expanded if a further study is ever undertaken.

(2) author's response As state in our conclusions, the C isotope results are still preliminary and need further understanding. Nevertheless, they are promising as a commercial traceability tool that can be improved based on trophic marker studies.

(1) comments from Referees The problem of obtaining fish samples with a known origin is not unique to this study and it is recommended that fish obtained from markets be treated with caution.

(2) author's response We agree with this remark.

(1) comments from Referees The value of this work lies in the fact that it can be extended to solve more than one issue. The biggest being the illegal use of protected natural fish populations. The techniques will work for other fish species as well and it will provide data as to the sediment load and erotional patterns of such large rivers.

(2) author's response We thank the reviewer for these positive points of view, which we share!

(3) author's changes in manuscript. Editions:

Page 2 line 9: replace "certificate" with "characterize" Page 3 line 17: replace with: "The Amazon basin represents a dynamic and heterogeneous ecosystem extending over more the 45% of the surface area of South America." Page 3 line 18: replace: "geologic" with "geological" Page 3 line 20: replace with "These habitats are therefore some of the most biodiverse in the world, particularly in regard to the Amazonian freshwater fish fauna which is under pressure of degradation by dams, buildings, mining, land cover and global climate change" Page 6 line 12: replace "Inter laboratorian" with "Inter-laboratory" Page 6 line 13: replace "repeatability" with "repeatability" Page 7 Figure 2: replace "87/86Sr" with "$^{87}Sr/^{86}Sr$" (numbers in superscript) and use this notation consistent throughout the entire document. Page 7 line 10: replace "The slice preparations were drilled per 6.8 mm interval" with "The slice preparations were drilled a 6.8 mm intervals" Page 7 line 18: replace "Also a test t was applied" with "A t-test

was also applied" Page 9 Figure 4: replace "87/86Sr" with "87Sr/86Sr" (numbers in superscript) and use this notation consistent throughout the entire document. Insert "87Sr/86Sr" (numbers in superscript) for part (b) of the figure also. Page 9 line 8: replace "interindividual" with "inter-individual" Page 10 Figure 5: replace "87/86Sr" with "87Sr/86Sr" (numbers in superscript) and use this notation consistent throughout the entire document. Insert "87Sr/86Sr" (numbers in superscript) for Central Amazon part of the figure also. Page 11 Figure 6: replace "87Sr/86Sr" with "87Sr/86Sr" (numbers in superscript) Page 11 line 16: replace "(sample origin in a row, predicted origin in the column)" with "(sample origin rows predicted origin columns)" Page 12 line 4: replace "(sample origin in a row, predicted origin in the column)" with "(sample origin rows, predicted origin columns)" Page 12 line 19: meaning unclear, please re-write sentence: "The role of food is more controversial as revealed by Sturrock et al. (2012) that reviewed significant or nonsignificant food up taking processes on the Sr isotopic composition of fish." Page 13 line 21: please confirm: "black water" or backwater" and be consistent throughout document.

We have accepted all of Jordaan's editing suggestion.

---

## Author Comment (AC3) · 8 Feb 2019

The authors revised the manuscript corrections and rectified the indication of pages and lines changed. Editions:

Page 2 line 10: replaced "certificate" with "characterize"; Page 4 line 3: replaced with: "The Amazon basin represents a dynamic and heterogeneous ecosystem extending over more the 45% of the surface area of South America."; Page 4 line 12: replaced: "geologic" with "geological"; Page 4 line 6: replaced with "These habitats are therefore some of the most biodiverse in the world, particularly in regard to the Amazonian fresh-water fish fauna which are under pressure of degradation by dams, buildings, mining,

land cover and global climate change" Page 8 line 9: replaced "Inter laboratorian" with "Inter-laboratory"; Page 8 line 21: replaced "repeatability" with "repeatability"; Page 9 Figure 2: replaced "87/86Sr" with "87Sr/86Sr" (numbers in superscript) and use this notation consistent throughout the entire document; Page 9 line 7: replace "The slice preparations were drilled per 6.8 mm interval" with "The slice preparations were drilled a 6.8 mm intervals"; Page 9 line 14: replaced "Also a test t was applied" with "A t-test was also applied" Page 10 Figure 4: replaced "87/86Sr" with "87Sr/86Sr" (numbers in superscript) and use this notation consistent throughout the entire document. Insert "87Sr/86Sr" (numbers in superscript) for part (b) of the figure also; Page 11 line 8: replaced "interindividual" with "inter-individual"; Page 11 Figure 5: replaced "87/86Sr" with "87Sr/86Sr" (numbers in superscript) and use this notation consistent throughout the entire document. Insert "87Sr/86Sr" (numbers in superscript) for Central Amazon part of the figure also; Page 12 Figure 6: replaced "87Sr/86Sr" with "87Sr/86Sr" (numbers in superscript); Page 13 line16: replace "(sample origin in a row, predicted origin in the column)" with "(sample origin rows, predicted origin columns)"; Page 14 line 4: replace "(sample origin in a row, predicted origin in the column)" with "(sample origin rows, predicted origin columns)"; Page 14 line 18: meaning unclear, please re-write sentence: "The role of food is more controversial as revealed by Sturrock et al. (2012) that reviewed significant or nonsignificant food up taking processes on the Sr isotopic composition of fish."; Page 15 line 21: please confirm: "black water" or backwater" and be consistent throughout document.; We thank the reviewer for the suggestions.

---

## Author Comment (AC4) · 8 Feb 2019

The authors revised the manuscript and rectified the indications of pages and lines in the text.

page 2 lines 26-29: We have included a better description of the existing contrast in Sr isotopic composition among the different sub-basins of the Amazon, to provide a clear background for the reader; page 3 lines 8-15: We have addressed this issue in the introduction section of the revised manuscript as it constitutes an important justification of the study; page 3 lines 16-26: We have revised it and included a description of arapaima habitat and trophic relations; pages 4, line 31- page 5 line 4: We have also

emphasized the lack of precise information about fish capture location, which is part of the fragility of the actual system of traceability; page 14, lines 09-10: We have cited previous studies about arapaima migration; page 14, lines 10-11; page 17 line 15-20: We have stated that the lack of precise information of fish origin have limited the approach and are one of the main causes of uncertainty.

We strongly valued all the suggestions made that are important contributions to improve the quality of the work.

---

## Author Response (AR1)

**Authors' response**

**(1) comments from Referees**

The manuscript is well written, has a proper and logical layout and makes a valid and valuable scientific contribution. The text can, however, benefit from minor editing (see notes below under editing). The authors investigated a very complex river system draining an even more complex geological setting. Sr isotope analyses is a good method for investigating such a system as it is mainly determined by the weathering of up-stream geology which is not expected to vary much within a 10 or even 50 years period. Please see Jordaan, L.J., Wepener, W. and Huizenga, J.M. (2016). The strontium isotope distribution in water and fish within C1 BGD Interactive comment Printer-friendly version Discussion paper major South African catchments. Water SA, 42(2), 213-224. In this case, very similar data were obtained in a smaller and more controlled river system, but it confirms the underlying assumptions made by the authors for the Amazon system. Data gathering can be much expanded over a multi-year sampling period to include seasonal variation if a further study is ever undertaken.

**(2) author's response**

We fully agree that it is important to monitor river seasonal variation of Sr isotope to improve the Amazon system knowledge. The Hybam project has been providing pioneer monitoring and characterization of the Amazon system, including monthly water sampling for more than 10 years at 15 sites (Santos et al. 2015). This data set constitutes a baseline and background that has benefited several studies related to fishery sciences (migration pathways, fish stock localization, or like in our manuscript commercial traceability) as well as to the evolution of the basin. These data clearly highlight the Sr isotopic composition contrast among major Amazon sub-basins as well as seasonal isotopic fluctuations related to differential erosion sources (see Santos et al. 2015). Due to the logistic complexity of water sampling in the Amazon system, it is difficult and expensive to expand the monitoring although it is surely an important issue. We hope that our work to provide a new application of this knowledge, thus reinforcing the argument to improve such a monitoring program. Our finds are in agreement with Jordaan et al. (2016), which will be included in our revised manuscript. In the Amazon basin, Pouilly et al. (2014) established the correlation between Sr isotopic ratio in water and in fish, as provided by Jordaan et al. (2016) in South Africa.

**(1) comments from Referees**

Using C isotopes is a good approach to this problem. It has however several more factors influencing isotope fractionation than Sr isotopes and should be used within the constraints of the technique, as the authors rightfully did. Data can be much expanded if further study is ever undertaken.

**(2) author's response**

As state in our conclusions, the C isotope results are still preliminary and need further understanding. Nevertheless, they are promising as a commercial traceability tool that can be improved based on trophic marker studies.

**(1) comments from Referees**

The problem of obtaining fish samples with a known origin is not unique to this study and it is recommended that fish obtained from markets be treated with caution.

**(2) author's response**

We agree with this remark.

**(1) comments from Referees**

The value of this work lies in the fact that it can be extended to solve more than one issue. The biggest being the illegal use of protected natural fish populations. The techniques will work for other fish species as well and it will provide data as to the sediment load and erotional patterns of such large rivers.

**(2) author's response**

We thank the reviewer for these positive points of view, which we share!

**(1) comments from Referees #2**

I have reviewed this ms and find it to deserve publication after a few revisions are made. The ms tests the general idea that biochemical tags can be used to identify the origin of harvested individuals of arapaima in various regions of the Amazon, and this can be used to improve the management of this economically important but overexploited fish. While the authors have done an apparent good job in analyzing data, I feel like the true contribution of this ms is not reflected in the text. The introduction generally sets out the research question clearly, but there are important issues that were not considered and would help sharpen it and increase the value of this research. For instance, about 3/4 of the introduction is devoted to describing the use of biochemical tags to trace the origin of fish worldwide, and Amazon is introduced only after that. When the subject of the Amazon is introduced, a key idea that is missing here is spatial heterogeneity in the chemistry of river waters. That heterogeneity is what allows the authors to test the main hypothesis, yet it is not described here, not even briefly. The hypothesis only makes sense IF there is spatial heterogeneity in chemistry, so this needs to be established in the introduction (and could be expanded in methods).

**(2) author's response**

We thank the referee for this important comment. We have included a better description of the existing contrasts in Sr isotopic composition among the different sub-basins of the Amazon, to provide a clear background for the reader (page 2 lines 28-30)

**(1) comments from Referees**

Also, the study focuses on arapaima and its conservation. But while a lot of space in the introduction is devoted to reviewing the use of chemical markers for sustainable fisheries management in general terms (paragraphs 1-3), there is almost no mention of details about the conservation measures that currently are bringing arapaima back from overexploitation. Given the paper focuses on arapaima, that seems to need attention. In particular, some 500 fishing communities in the State of Amazonas in Brazil (alone, and more now in Para State) are setting fishing quotas and selling their "sustainable" fish to the market while fulfilling strict government limits. Each individual fish harvested under this management system receives a unique, government-issued, identifying a tag that buyers can use to know where the fish came from, where and when it was harvested. But many such tags are illegally re-used to allow the "legal" sale of unsustainably harvested fish. This presents a major management problem for arapaima that the study in question can help solve because its results can potentially be used to 'trace' back the origin of the fish and hence determine if the origin of the fish matches the tag. This study should link its results to such major ongoing management initiative for arapaima.

**(2) author's response**

We thank the referee for this important comment and for the details. We have addressed this issue in the introduction section of the revised manuscript as it constitutes an important justification of the study (page 3 lines 8-15). Since 1989 Arapaima stocks are recovering, indicating that the adaptive management of Arapaima fisheries has been a success and may hopefully became a positive example of synergetic social and political actions in the region. However, as stated by the referee the situation is not completely controlled and we hope the development of such tools of commercial traceability would help to improve further the situation.

**(1) comments from Referees**

Finally, the hypothesis of the study only makes sense IF arapaima is not highly migratory and move between and among river systems with different water chemistries. As such, known data on the general migratory behavior of arapaima should be determined 'before' the hypothesis for the hypothesis to make sense. When this is done, typical habitat and food sources (some of which are presented in methods), should also be presented here, to provide context for the hypothesis. To

implement such changes, I suggest shortening the first 3 paragraphs and expanding the remainder of the introduction.

**(2) author's response**

We agree with the reviewer comments and suggestion about the introduction. We have revised it and included a description of arapaima habitat and trophic relations (page 3 lines 21-27).

**(1) comments from Referees**

Fig 1 needs to be edited so the font is readable at half page width size; currently, the most important information cannot be read for having font too small. A lot of space on the sides is used to show regions of no interest. The fig could be "zoomed in" to the area of interest. As for the analysis, there is a major mismatch in geographical precision of the otolith data. The fish otoliths were collected from fishermen residing in sites (Table 1). Each of the 'sites' mentioned, such as Itacoatiara, Manaus, or Mamiraua ˛A reserves are, in fact, enormous regions, each of which encompasses several different habitats, each of which can have varied water chemistries. For instance, the Mamiraua Reserve is flooded by some five or more major river tributaries and includes surrounding areas influenced by blackwater ria lakes. This variability in regional water chemistry is not matched by the literature data used for each "site" It also constitutes a limitation of the analytical approach undertaken. More detail on fish capture location will be introduced Could this lack of specificity in the otolith origin data help explain part of the unexplained variance in the analyses? I would seem so. As such, the "match" between the otolith and water chemistry data should be presented and discussed in methods, as well as in the discussion. It does not invalidate the analysis, but it adds more nuance and probably helps explain its results.

**(2) author's response**

We have improved Fig. 1 by zooming and increasing the fonts. We have also emphasized the lack of precise information about fish capture location, which is part of the fragility of the actual system of traceability (page 5, lines 1-10). We agree that this issue, which may be related to the unexplained variance of the analysis and have added another row in Table. 1 about the specimen origin. This topic was further addressed in the discussion sections.

**(1) comments from Referees**

Discussion: Line 10: are there movement studies showing arapaima do not migrate long distances? If so, this is the place to cite them (again, after introducing them in the intro)

**(2)  author's response**

We have cited previous studies about arapaima migration (page 14, lines 9-10).

**(1)  comments from Referees**

Lines 12-14: this is where a well-developed discussion of the potential for lack of geographical specificity in the otolith data to influence the results could go.

**(2)  author's response**

We have stated that the lack of precise information of fish origin have limited the approach and are one of the main causes of incongruity in the analysis (page 14, lines 10; page 18 line 19-21)

**(1)  comments from Referees**

In general, the text of the discussion is sound. But I find it to be too long and unclear at times, so I suggest condensing it and revising it for clarity. What is really missing is linking the results to their application, following the idea suggested above.

**(2)  author's response**

We have revised the text in order to make it clearer.

**(1)  comments from Referees**

Spp is not italicized Line 25: use 'developed' instead of 'satisfying'

**(2)  author's response**

We have made the changes suggested by the reviewer. The authors strongly valued all the suggestions made that are important contributions to improve the quality of the work.

We strongly valued all the suggestions made that are important contributions to improve the quality of the work.

**(3) Editions**

- Over all the documented spp was formatted as non-italic
- The in text citation and references was reviewed.
- The authors' filiation was reviewed
- Page 1 line 25 replaced satisfying with developed
- The introduction section was restructured according to the Referee 2 suggestions. the review of biogeochemical tag was shortened and a better description of the method assumption was provided as well as a contextualization of actual system of management and traceability of Arapaima spp.
- Page 2 line 10: replaced "certificate" with "characterize";
- page 2 lines 28-30: We have included a better description of the existing contrast in Sr isotopic composition among the different sub-basins of the Amazon, to provide a clear background for the reader;
- page 3 lines 8-15: We have addressed this issue in the introduction section of the revised manuscript as it constitutes an important justification of the study;
- page 3 lines 21-27: We have revised it and included a description of arapaima habitat and trophic relations;
- Page 4 line 5: replaced with: "The Amazon basin represents a dynamic and heterogeneous ecosystem extending over more the 45% of the surface area of South America.";
- Page 4 line 8: replaced with "These habitats are therefore some of the most biodiverse in the world, particularly in regard to the Amazonian freshwater fish fauna which is under pressure of degradation by dams, buildings, mining, land cover and global climate change"
- Page 4 line 12: replaced: "geologic" with "geological";
- page 5 Fig1-zoomed and the font was increased
- page 5, lines 1-10: We have also emphasized the lack of precise information about fish capture location, which is part of the fragility of the actual system of traceability;
- page 5-6 we added a column in table 1 with the informed origin of the fishes and reviewed the Mean Water $^{87}Sr/^{86}Sr$
- Page 7 line 9: replaced "Inter laboratorian" with "Inter-laboratory";
- Page 8 line 10: replaced "repitability" with "repeatability";
- Page 8 Figure 2: replaced "87/86Sr" with "$^{87}Sr/^{86}Sr$" (numbers in superscript) and use this notation consistent throughout the entire document;
- Page 9 line 2: replace "The slice preparations were drilled per 6.8 mm interval" with "The slice preparations were drilled with intervals of 6.8 mm"
- Page 9 line 10: replaced "Also a test t was applied" with "A t-test was also applied"
- Page 11 Figure 4: replaced "87/86Sr" with "$^{87}Sr/^{86}Sr$" (numbers in superscript) and use this notation consistent throughout the entire document. Inserted "$^{87}Sr/^{86}Sr$" (numbers in superscript) for part (b) of the figure also;
- Page 11 line 8: replaced "interindividual" with "inter-individual";
- Page 12 Figure 5: replaced "87/86Sr" with "$^{87}Sr/^{86}Sr$" (numbers in superscript) and used this notation consistent throughout the entire document. Insert "$^{87}Sr/^{86}Sr$" (numbers in superscript) for Central Amazon part of the figure also;
- Page 13 Figure 6: replaced "87Sr/86Sr" with "$^{87}Sr/^{86}Sr$" (numbers in superscript);

- Page 13 line14: replace "(sample origin in a row, predicted origin in the column)" with "(sample origin rows, predicted origin columns)";
- Page 14 line 1: replace "(sample origin in a row, predicted origin in the column)" with "(sample origin rows, predicted origin columns)";
- page 14, lines 9-10: We have cited previous studies about arapaima migration;
- Page 14 line 18: changed: "The role of food is more controversial as revealed by Sturrock et al. (2012) that reviewed significant or nonsignificant food up taking processes on the Sr isotopic composition of fish." Per "the role of food in the strontium otolith uptake is debated, as revealed by Sturrock et al. (2012) that reviewed the significance of food up taking processes on the Sr isotopic composition of fish"
- page 14, lines 10; page 18 line 19-21: We have stated that the lack of precise information of fish origin have limited the approach and are one of the main causes of uncertainty.
- Page 15 line 21: confirmed: "backwater" throughout the document.

**(4) Marked-up manuscript version**

[revised manuscript text omitted]

Formatado: Francês (França)

Formatado: Francês (França)

Formatado: Francês (França)

Formatado: Francês (França)

Formatado: Francês (França)

The carbon isotopic composition of an organism depends primarily on the isotopic composition of the primary producer that constitutes the beginning of the trophic chain it feeds on. In particular, plants present two main photosynthetic pathways, C3 and C4, which produce a two-contrasted $\delta^{13}$C range of values (-32‰ to -24‰ for C3 plants; -14‰ to -9‰ for C4 plants; De Niro & Epstein, 1978). In general, wild Amazon fishes feed dominantly on a trophic chain derived from C3 carbon source (Araujo-Lima et al., 1986; Forsberg et al., 1993; Jepsen and Winemiller, 2007; Marshall et al., 2008; Mortillaro et al., 2015; Watson et al., 2013).

**2.2 Fish otolith sampling**

Thirty-eight otolith samples of *Arapaima* spp. were collected in different sites of four main regions (Figure 1): 22 were obtained from professional fishermen for a commercial purpose; 6 were obtained from farmers, and 10 others were collected in Manaus and Santarém markets (Table 1). The sagittae otoliths of each specimen were extracted by head dissection after capture. Afterward, they were washed, dried, and kept until laboratory analyses. None of the fishes had precise location of capture, and the informed origin in four main regions were (Figure 1): Solimões (Mamirauá Reserve at the Solimões river), Madeira (Yata, Beni and Madre de Dios Rivers and Ariquemes farm), Central Amazon (Manacapuru farm, at Solimoes River; Itacoatiara, at Amazon River; Novo Airao at Negro River) and Lower Amazon (Santarem, Amazon river). These regions have a complex drainage system that may include different sources of water with different Sr isotopic compositions in each site (Table 1). For example, Mamiraua Reserve in Solimoes region is located at the confluence of five major rivers, which waters range from white (lower $^{87}$Sr/$^{86}$Sr values) to black water type (higher $^{87}$Sr/$^{86}$Sr values). Similarly, Itacoatiara is located at the confluence zone of the Amazon and Madeira rivers, which also have quite distinct Sr isotopic compositions. Therefore, the lack of precise information of collection and the variability in regional water chemistry may not be exactly matched by literature data and may be important to explain the unexpected variance of fish otolith data.

[Figure]

[Figure]

**Figure 1. Map of the Amazon basin showing the regions and sampling sites for *Arapaima spp*. Wild collect sites are represented by red squares, farm collect sites by blue squares. The Amazon River (Lower Amazon and Central Amazon) is colored blue, the Solimões green, the Madeira purple, and the Negro  red.**

**Table 1. $\delta^{13}C$ (mean ±SD) and $^{87}Sr/^{86}Sr$ (mean ±SD) measured in sagittae otolith for 38 wild and farmed *Arapaima gigas* proceeding from four Amazonian regions with correct classification and correspondent water $^{87}Sr/^{86}Sr$ reviewed from literature. [1]Palmer and Edmond (1992), [2]Gaillardet et al. (1997), [3]Queiroz et al. (2009), [4]Pouilly et al. (2014), [5]Santos et al. (2015) and [6]Duponchelle et al. (2016)**

| Specimen code |  | Fish origin | Site of collection | Informed origin | Mean Water $^{87}Sr/^{86}Sr$ | | Otolith $^{87}Sr/^{86}Sr$ | QDA prediction  | Otolith $\delta^{13}C$ |
|---|---|---|---|---|---|---|---|---|---|
| COO |  | Central Amazon | Farming | Farming | Manacapuru  |  | 0.091±0.[2,3][5,6] | 0.71543 | Central Am | -18.6 |
| I1 |  | Central Amazon | Wild | Fisherman | Itacoatiara  |  | 0.7111 ±0.0004[2,5,6] | 0.71029 | Central | -18.3 |

Amazon Am

| ID | | Region | Management | Market | Location | | Value ±error | | Region | δ |
|---|---|---|---|---|---|---|---|---|---|---|
| I2 | Wild | Central Amazon | Wild | Fishermen | Itacoatiara fishers | 0.71267 | 0.7111 ±0.0004[2,6] | 0.71267 | Central Amazon Am | -15.4 |
| I3 | Wild | Central Amazon | Wild | Fishermen | Itacoatiara fishers | 0.71020 | 0.7111 ±0.0004[2,6] | 0.71020 | Central Amazon Am | -20.0 |
| I4 | Wild | Central Amazon | Wild | Fishermen | Itacoatiara fishers | 0.71041 | 0.7111 ±0.0004[2,6] | 0.71041 | Central Amazon Am | -19.9 |
| NA1 | Farming | Central Amazon | Farming | Manaus market | Novo Airao 0.70997 | | 0.71117300 ±0.00040138[2,6] | 0.70997 | Solimões | -25.2 |
| NA2 | Farming | Central Amazon | Farming | Manaus market | Novo Airao | 0.7300 ±0.0138[2] | 0.70977 | 0.7111 ±0.0004[2,6] | Solimões | -25.2 |
| NA3 | Farming | Central Amazon | Farming | Manaus market | Novo Airao | 0.7300 ±0.0138[2] | 0.70926 | 0.7111 ±0.0004[2,6] | Solimões | -26.0 |
| NA4 | Farming | Central Amazon | Farming | Manaus market | Novo Airao | 0.7300 ±0.0138[2] | 0.70982 | 0.7111 ±0.0004[2,6] | Solimões | -24.1 |
| FMn1 | Farming | Solimões | Farming | Manaus market | Tefé 0.70896 | 0.70896 | 0.7095±0.0008[2,3,6] | 0.70896 | Solimões | -25.1 |
| FMn2 | Farming | Solimões | Farming | Manaus market | Tefé 0.70894 | 0.70894 | 0.7095±0.0008[2,3,6] | 0.70894 | Solimões | -24.1 |
| FMn3 | Farming | Solimões | Farming | Manaus market | Tefé 0.70894 | 0.70894 | 0.7095±0.0008[2,3,6] | 0.70894 | Solimões | -23.2 |
| M1 | Wild | Solimões | Wild | Fishermen | Mamirauá Reserve | 0.70904 | 0.7095±0.0008[2,3,6] | 0.70904 | Solimões | -25.6 |
| M2 | Wild | Solimões | Wild | Fishermen | Mamirauá Reserve | 0.70932 | 0.7095±0.0008[2,3,6] | 0.70932 | Solimões | -25.4 |

**Células Inseridas**
**Células Inseridas**
**Células Excluídas**
**Células Inseridas**
**Células Inseridas**
**Células Excluídas**

| | | | | | | | | | | |
|---|---|---|---|---|---|---|---|---|---|---|
| M3 |  | Solimões | Wild | Fisherme n | Mamir auá Reserv e |  | 0.7095±0.0008 [2,3,6] | 0.70963 | Solimõ es | -24.8 |
| M4 |  | Solimões | Wild | Fisherme n | Mamir auá Reserv e |  | 0.7095±0.0008 [2,3,6] | 0.70941 | Solimõ es | -25.1 |
| M5 |  | Solimões | Wild | Fisherme n | Mamir auá Reserv e |  | 0.7095±0.0008 [2,3,6] | 0.70944 | Solimõ es | -25.1 |
| CS1 |  | Lower Amazon | Farming | Santarem market | Santarem | | 0.7112±0.0004 [2,3,6] | 0.71382 | Lower Am | -13.7 |
| CS2 |  | Lower Amazon | Farming | Santarem market | Santarem | | 0.7112±0.0004 [2,3] | 0.71291 | Lower Am | -15.6 |
| CS5 |  | Lower Amazon |  | antarem arket | Santarem | | 0.7112±0.0004[2,3,6] 0.71479 | | Lower Am | -14.0 |
| S1 |  | Lower Amazon | Wild | Fisherme n | Santarem | | 0.7112±0.0004 [2,3,6] | 0.70868 | Solim ões | -22.9 |
| S3 |  | Lower Amazon | Wild | Fisherme n | Santarem | | 0.7112±0.0004 | 0.70873 | Solim | |
| S4 |  | Lower Amazon | Wild | Fisherme n | Santarem | | 0.7112±0.0004 [2,3,6] | 0.70868 | Solim ões | -28.4 |
| S5 |  | Lower Amazon | Wild | Fisherme n | Santarem | | 0.7112±0.0004 [2,3,6] | 0.70865 | Solim ões | -25.6 |
| R1 | | Madeira | Farming | F arming | Arique mes  |  | 0.7188±0.0012 [1,5,6] | 0.72245 | Madei ra | -8.7 |
| R2 | | Madeira | Farming | F arming | Arique mes  |  | 0.7188±0.0012 [1,5,6] | 0.71479 | Madei ra | -8.3 |
| R3 | | Madeira | Farming | F arming | Arique mes  |  | 0.7188±0.0012 [1,5,6] | 0.71479 | Madei ra | -4.8 |

**Células Inseridas**

**Formatado**

**Formatado**

**Formatado**

**Formatado**

**Formatado**

**Formatado**

**Formatado**

**Formatado**

**Células Inseridas**

**Células Excluídas**

| Sample | Type | | | Location | | Value | | | | Madeira | Temp |
|---|---|---|---|---|---|---|---|---|---|---|---|
| R4 | Madeira | Farming | MadeiraFarming | Ariquemes farming | 0.71479 | 0.7188±0.0012 [1,5,6] | 0.71479 | | | Madeira | -8.6 |
| R5 | Madeira | Farming | Farming Madeira | Ariquemes farming | 0.71479 | 0.7188±0.0012 [1,5,6] | 0.71479 | | | Madeira | -8.2 |
| YA-1 | Wild Madeira | Wild | Fishermen | Yata fishers river | 0.72854 | 0.7245±0.0018 [4,6] | 0.72854 | | | Madeira | -24.8 |
| YA-12 | Wild Madeira | Wild | Fishermen | Yata fishers river | 0.72848 | 0.7245±0.0018 [4,6] | 0.72848 | | | Madeira | -26.6 |
| YA-13 | Wild Madeira | Wild | Fishermen | Yata fishers river | 0.72885 | 0.7245±0.0018 [4,6] | 0.72885 | | | Madeira | -28.1 |
| FL-1 | Wild Madeira | WildMadre de Dios fishers | Fishermen | Beni river | 0.7184±0.0011 [1,6] | 0.72468 | 0.7100±0.0004 [1,6] | | | Madeira | -27.6 |
| FL-2 | Wild Madeira | WildMadre de Dios fishers | Fishermen | Beni river | 0.7184±0.0011 [1,6] | 0.72143 | 0.7100±0.0004 [1,6] | | | Madeira | -27.0 |
| FL-20 | Wild Madeira | WildMadre de Dios fishers | Fishermen | Beni river | 0.7184±0.0011 [1,6] | 0.72232 | 0.7100±0.0004 [1,6] | | | Madeira | -24.8 |
| LV-6 | Wild Madeira | Wild | Fishermen | Madre de Dios fishers | 0.71441 | 0.71007119±0.0004 [1,6] | 0.71441 | | | Madeira | -24.7 |
| LV-18 | Wild Madeira | Wild | Fishermen | Madre de Dios fishers | 0.71479 | 0.71007119±0.0004 [1,6] | 0.71479 | | | Madeira | -23.7 |
| LV-20 | Wild Madeira | Wild | Fishermen | Madre de Dios fishers | 0.72232 | 0.71007119±0.0004 [1,6] | 0.72232 | | | Madeira | -23.8 |

Additionally, three samples of farmed fish food commonly used to feed Arapaima were obtained from farmers. The three food types used to feed the different fishes life's stages were collected,

Formatado: Recuo: Primeira linha: 0,01 cm
Formatado: Recuo: Primeira linha: 0,01 cm
Células Excluídas
Células Inseridas
Células Inseridas
Formatado: Recuo: Primeira linha: 0,04 cm, À direita: -0,26 cm
Formatado: Recuo: Primeira linha: 0,01 cm
Formatado
Formatado: Recuo: Primeira linha: 0,01 cm
Formatado
Formatado: Recuo: Primeira linha: 0,01 cm
Células Excluídas
Formatado
Formatado: Recuo: Primeira linha: 0,04 cm
Células Inseridas
Células Inseridas
Formatado
Formatado: Recuo: Primeira linha: 0,04 cm
Formatado
Formatado: Recuo: Primeira linha: 0,04 cm
Formatado
Células Inseridas
Formatado: Português (Brasil)
Formatado: Recuo: Primeira linha: 0,01 cm
Células Excluídas
Células Inseridas
Formatado: Português (Brasil)
Formatado
Formatado: Português (Brasil)
Formatado: Recuo: Primeira linha: 0,01 cm
Formatado: Português (Brasil)
Formatado
Formatado: Português (Brasil)
Formatado: Recuo: Primeira linha: 0,01 cm
Formatado: Inglês (Estados Unidos)
Formatado: Português (Brasil)
Formatado: Recuo: Primeira linha: 0 cm

[revised manuscript text omitted]